# An in silico molecular docking and simulation study to identify potential anticancer phytochemicals targeting the RAS signaling pathway

Mahir Azmal, Jibon Kumar Paul, Fatema Sultana Prima, Omar Faruk Talukder, Ajit Ghosh*

Department of Biochemistry and Molecular Biology, Shahjalal University of Science and Technology, Sylhet, Bangladesh

* aghosh-bmb@sust.edu

## Abstract

The dysregulation of the rat sarcoma (RAS) signaling pathway, particularly the MAPK/ERK cascade, is a hallmark of many cancers, leading to uncontrolled cellular proliferation and resistance to apoptosis-inducing treatments. Dysregulation of the MAPK/ERK pathway is common in various cancers including pancreatic, lung, and colon cancers, making it a critical target for therapeutic intervention. Natural compounds, especially phytochemicals, offer a promising avenue for developing new anticancer therapies due to their potential to interfere with these signaling pathways. This study investigates the potential of anticancer phytochemicals to inhibit the MAPK/ERK pathway through molecular docking and simulation techniques. A total of 26 phytochemicals were screened from an initial set of 340 phytochemicals which were retrieved from Dr. Duke's database using in silico methods for their binding affinity and stability. Molecular docking was performed to identify key interactions with ERK2, followed by molecular dynamics (MD) simulations to evaluate the stability of these interactions. The study identified several phytochemicals, including luteolin, hispidulin, and isorhamnetin with a binding score of -10.1±0 Kcal/mol, -9.86±0.15 Kcal/mol, -9.76 ±0.025 Kcal/mol, respectively as promising inhibitors of the ERK2 protein. These compounds demonstrated significant binding affinities and stable interactions with ERK2 in MD simulation studies up to 200ns, particularly at the active site. The radius of gyration analysis confirmed the stability of these phytochemical-protein complexes' compactness, indicating their potential to inhibit ERK activity. The stability and binding affinity of these compounds suggest that they can effectively inhibit ERK2 activity, potentially leading to more effective and less toxic cancer treatments. The findings underscore the therapeutic promise of these phytochemicals, which could serve as a basis for developing new cancer therapies.

**Data Availability Statement:** All relevant data are within the manuscript and its Supporting Information files.

**Funding:** AG has received partial funding from the Shahjalal University of Science and Technology Research Center (LS/2023/1/01).

**Competing interests:** The authors have declared that no competing interests exist.

**Abbreviations:** RTK, Receptor Tyrosine Kinase; CRAF, C-Rapidly Accelerated Fibrosarcoma (also known as RAF1); MEK, Mitogen-activated protein kinase kinase (MAP2K); BRAF, B-Rapidly Accelerated Fibrosarcoma; BRAFi, BRAF inhibitors; ERK, Extracellular signal-regulated kinase; MEKi, MEK inhibitors; ERKi, ERK inhibitors; c-JUN, c-Jun proto-oncogene; c-FOS, c-Fos proto-oncogene; ELK, E26 transformation-specific (ETS)-like transcription factor; ETS, E26 transformation-specific transcription factors; Cyclin D1, Cyclin D1 protein; BIM, Bcl-2-like protein 11; MCL, Myeloid cell leukemia sequence; RSK, Ribosomal S6 kinase.

## Introduction

The mitogen-activated protein kinase (MAPK) pathway facilitates the transmission of extracellular signals from the cell membrane to intracellular locations and plays a role in diverse biological processes [1]. It is highly conserved across eukaryotic organisms and is fundamental in transducing extracellular signals into intracellular responses [2]. The dysregulation of the MAPK pathway has been observed in numerous renin-angiotensin system (RAS)-associated malignancies. Mutations in the RAS gene give rise to the persistent activation of the MAPK pathway, leading to unregulated cellular proliferation and the development of resistance to medications that induce apoptosis [3–5]. The potential therapy approach for RAS-driven tumors involves disrupting signals between the RAS and downstream effectors, specifically the RAF-MAPK kinase (MEK)–extracellular signal-related kinase (ERK) pathway [6–8]. ERKs are part of the mitogen-activated protein kinase (MAPK) family, including c-Jun N-terminal kinases (JNKs) and p38 MAPKs. ERKs are activated in response to extracellular signals such as growth factors, hormones, and cytokines. Upon stimulation, a cascade of phosphorylation events is initiated, leading to the activation of ERKs. Once activated, ERKs translocate to the nucleus, phosphorylating transcription factors and other nuclear targets, thereby regulating gene expression and influencing cellular responses [9, 10]. Dysregulation or hyperactivation of the ERK pathway has been associated with various types of cancer, including melanoma, colorectal cancer, pancreatic cancer, and lung cancer. This pathway controls cell proliferation, survival, and differentiation, and aberrant activation of ERK signaling can lead to uncontrolled cell growth and tumor formation. Therefore, targeting components of the ERK pathway has become a promising strategy for cancer therapy [11, 12].

ERK exists in two main isoforms: ERK1 (p44 MAPK) and ERK2 (p42 MAPK), which share high sequence similarity and are often referred to collectively as ERK1/2. These isoforms are highly conserved among species, indicating their fundamental importance in cellular function [13]. ERK protein plays a central player in the RAF/MEK/ERK signaling cascade, relaying extracellular signals to the nucleus to regulate essential cellular functions critical for normal development and homeostasis [14]. The MAPK/ERK pathway, specifically focusing on components like BRAF or MEK, has emerged as a prominent area of interest in cancer treatment.

Novel small molecule inhibitors targeting these constituents have been formulated and authorized to manage particular malignancies, including melanoma [4, 5, 14]. Plants offer a vast array of natural resources. The report highlights the importance of natural goods in healthcare, revealing that 80% of the world's population depends on plant-based medications to meet their healthcare requirements [15, 16]. The development of phytochemical-based therapies holds promise for combating RAS-driven malignancies and other cancers characterized by dysregulated MAPK/ERK signaling. These compounds offer several advantages over conventional chemotherapeutic agents, including their relatively low toxicity, high bioavailability, and pleiotropic effects on cancer cells [17, 18]. The impact of several phytochemicals, with a specific focus on flavonoids, polyphenolic compounds also some terpenoids, on protein kinases, specifically in the inhibition of signal transduction pathways (e.g., MAPK/ERK pathway), induction of apoptosis through modulation of pro-apoptotic and anti-apoptotic proteins, antioxidant activity reducing oxidative stress, epigenetic modulation altering DNA methylation and histone acetylation, inhibition of angiogenesis by downregulating pro-angiogenic factors like VEGF, immune system modulation enhancing NK cells and macrophage activity within the framework of cancer therapy [19–21]. Some present studies have been identified *in vitro* and *in vivo* by modulating the autophagy-apoptosis pathway (i.e., sulforaphane, resveratrol, lycopene, epigallocatechin, curcumin, and berberine) are currently being investigated in clinical trials for different cancer types [22]. Another in silico study being analyzed in

recent years identified various phytochemicals such as epigallocatechin gallate, piperine, gingerol, and thymoquinone showed substantial binding with P53 and NOTCH proteins to act as potential agents against breast cancer [23].

This study underscores the significance of phytochemicals, particularly flavonoids, which have long been of interest due to their potential health benefits, including anticancer properties as potential therapeutic agents targeting the MAPK/ERK pathway in cancer treatment. Through molecular docking and simulation studies, we elucidated the interaction between phytochemicals and ERK protein, shedding light on their inhibitory effects on this crucial signaling pathway implicated in tumorigenesis. These findings provide valuable insights into the development of novel phytochemical-based therapies for combating RAS-driven malignancies and offer a promising avenue for further research in cancer treatment. Overall, this study represents a significant advancement in our understanding of how phytochemicals may modulate the ERK pathway and offers promising prospects for the development of innovative cancer treatments. It underscores the importance of further research in this area to validate these findings and translate them into clinically relevant therapies for improving cancer outcomes.

## Materials and methods

### Ligand selection

Phytochemicals with anticancer, anti-carcinomic, and cancer-preventive properties have been retrieved in the field of ligand prediction. A thorough examination of pertinent literature was undertaken to extract information regarding medicinal plants or their constituent phytochemicals that demonstrate characteristics suggestive of anti-cancer or cancer preventive or anticarcinomic activity. The botanical names of these plants were used as search terms in Dr. Duke's Phytochemical and Ethnobotanical Databases (https://phytochem.nal.usda.gov/). The acquired results were subsequently examined to identify potential phytochemicals with the mentioned characteristics. As there have been three biological activities taken for study, some similar compounds have been examined during the file compilation (S1 File). Moreover, PubChem (https://pubchem.ncbi.nlm.nih.gov/) was the sole database used for ligand searches, leading to the absence of Compound IDs (CIDs) for certain chemicals. Following a thorough selection process, the identified compounds were chosen for subsequent analysis. Afterward, the names of the chosen compounds were queried in the PubChem database, and their corresponding three-dimensional structures were obtained. For Positive control as potent ERK inhibitor Pubchem CID 135523966 N-[1-(3-Chloro-4-fluorophenyl)-2-hydroxyethyl]-3-[4-(3-chlorophenyl)-1,2-dihydro-3H-pyrazol-3-ylidene]-3H-pyrrole-5-carboxamide (Pyrazolylpyrrole) [24], and the known ligand Ulixertinib Pubchem (CID 11719003) [25, 26] and Ravoxertinib (Pubchem CID 71727581) [27, 28] were retrieved from the pubchem database.

### Protein selection

The structure of ERK in complex with a natural inhibitor (1TVO) was selected from RCSB PDB (https://www.rcsb.org) for the docking process to know more about the active side of that protein. The PDB resolution was 2.50 Å. The rationale for choosing 1TVO over other available ERK2 structures is based on several factors. The 2.50 Å resolution of 1TVO provides a clear and precise depiction of the active site and the interactions with the inhibitor, which is crucial for accurate docking studies. Many other ERK2 structures in the PDB database have missing residues in critical regions, which would require computational adjustments and might introduce inaccuracies. By selecting 1TVO, the study could be able to avoid the potential complications and uncertainties associated with modeling these missing residues.

## Active site prediction

The active region on the surface of the protein that performs protein function is known as a protein-ligand binding site. To avoid blind docking the specific amino acid residue of protein-ligand interaction was predicted using CASTP v3.0 (http://sts.bioe.uic.edu/castp/calculation.html). For validation of the active side, the RCSB PDB server was checked. The ligand interaction with 1TVO has been observed and cross-checked the residues with the CASTp given data.

## ADMET profiling

For the ADME profiling, the SwissADME (http://www.swissadme.ch/index.php) server was deployed. To perform ADME analysis canonical smiles of ligands were needed. For Toxicity profiling pkCSM (https://biosig.lab.uq.edu.au/pkcsm) server was used. All the ligands' canonical smiles were stored in a text document and were used as input on the pkCSM server. All the data was downloaded in CSV format and sorted further based on the following criteria (Table 1).

## Cross-docking analysis with co-crystal ligand

Cross-docking was performed by docking for the ligand bound to the target receptor protein. This was done to assess the binding versatility and to identify potential new binding modes. Root Mean Square Deviation (RMSD) values between the docked poses and the reference crystal structure were calculated using Biovia Discovery Studio. RMSD values were used to assess the accuracy of the docking predictions, with RMSD < 2.0 Å considered as near-native poses [29]. The binding interactions between the ligands and the receptor were visualized using Biovia Discovery Studio. Hydrogen bonds, hydrophobic interactions, and other non-covalent interactions were analyzed to understand the binding mode and interaction strength of each ligand.

## Molecular docking of phytochemical

The compounds that have successfully undergone ADMET profiling were chosen for molecular docking with the selected protein. The docking was performed by using PyRx and Autodok tool. The target ligands were obtained from the PubChem database, 3D structures, typically in SDF format, and perform energy minimization using the steepest descent algorithm with a universal force field (UFF) to optimize their geometry. The rotatable bonds were checked to allow flexibility during docking. The minimized structures were converted into a docking-compatible format such as PDBQT, which involves adding hydrogens, setting partial charges, and defining torsional degrees of freedom. The target protein was transformed into pdbqt format, and a grid box was established based on its active site. The grid box size was set to (x = 126, y = 122, z = 70 Å) with a center at coordinates (15.416, -0.324, 13.078) the energy range was set to 4, and the exhaustiveness was increased to 10 to explore a broader range of potential binding modes for the Autodock docking and the PyRx docking the grid box was defined to x = 61.73, y = 45.08, z = 65.44 Å to ensure sufficient space for ligand binding. The center of the grid box was set to coordinates (15.4143, -0.3242, 9.9533). The exhaustiveness parameter was set to 8. Hydrogen atoms were positioned on the polar regions and Kollman charges were incorporated during protein preparation. Afterward, the docking results were evaluated for binding affinity, and all the resulting docked conformations were saved in a pdbqt file [30]. The docking results were expressed as a negative score in units of kcal/mol, with a lower score indicating a higher binding affinity. Furthermore, docking analysis was also performed for the established ligand and the binding interaction has been recorded.

**Table 1. AMDET profiling parameters and shorting criteria.**

| Molecular Weight g/mol [Min-Max] | Rotatable Bond Count [Min-Max] | Heavy Atom Count [Min-Max] | H-Bond Donor Count [Min-Max] | H-Bond Acceptor Count [Min-Max] | GI absorption | Polar Area, [Angstrom sq] [Min-Max] | Complexity [Min-Max] | XLOGP [Min-Max] | Rules 5 Out of 5 | AMES toxicity | Hepatotoxicity | Skin sensitivity |
|---|---|---|---|---|---|---|---|---|---|---|---|---|
| 120–500 | 1–10 | 12–30 | 0–4 | 0–10 | High | 4.9–104 | 144–494 | 1–5 | 0 violation | NO | NO | NO |

## Decoy screening of phytochemicals

Ligands that demonstrated better binding affinities than the positive control were selected for decoy screening to assess their specificity. The SMILES strings of these ligands were first generated and then input into the DUDE server (https://dude.docking.org), which provided corresponding decoy molecules [31]. These decoys are structurally like active ligands but are designed to avoid specific interactions with the target protein. The decoy SMILES strings were downloaded in text format and converted into 3D molecular structures using Open Babel software, resulting in SDF files [32]. These 3D decoy structures were then docked with the target protein using the same docking protocols applied to the active ligands. The docking results, including binding affinities, were retrieved in CSV format. By comparing the binding affinities of the decoys to those of the active ligands, the specificity of the inhibitors was assessed, helping to validate the docking outcomes and ensure the reliability of the identified inhibitors.

## Visualization of result

The result was visualized using Biovia Discovery Studio Visualizer 2021 and PyMol. The output files, output.pdbqt, and macromolecule, were opened concurrently in the PyMol software. During the docking process, a total of 9 distinct conformations were generated. However, for analysis, only the conformations with a root mean square deviation (RMSD) of 0 were taken into consideration. The docking affinity was compared with the positive control and natural ligand of the ERK protein, and the top result was selected for the simulation process [33]. The ligand and protein formed a protein-ligand docking complex, which was saved in pdb format for subsequent analysis and generation of binding site figures.

## MD simulation

An MD simulation lasting 200 nanoseconds was performed using the GROningen Machine for Chemical Simulations (GROMACS) version 2020.6. The simulation utilized the TIP3 water model. The entire system was subjected to energetic minimization using the CHARMM36 all-atom force field, as described previously [34]. The systems were neutralized by the addition of $Na^+$ and $Cl^-$ ions. The system underwent energy minimization, followed by isothermal isochoric (NVT) equilibration and isobaric (NPT) equilibration. Following that, a production MD simulation with a duration of 200 nanoseconds was started. The analysis of the MD simulation data comprised the calculation of several parameters, namely the Root Mean Square Deviation (RMSD), Root Mean Square Fluctuation (RMSF), Radius of Gyration (Rg), Solvent Accessible Surface Area (SASA), and Hydrogen Bond analysis [35]. The ggplot2 package, available at (https://ggplot2.tidyverse.org/) was used in RStudio to create visualizations for each analysis.

## Visualization of simulation results

Simulation results were visualized and analyzed with the grace tool in the Linux operating system. The graph was displayed in.png format in the result section.

## Metabolic pathway analysis

Metabolic pathway analysis was performed by the Kyoto Encyclopedia of Genes and Genomes (KEGG, https://www.genome.jp/kegg/pathway.html). The KEGG pathway is a collection of manually drawn pathway maps representing our knowledge of the molecular interaction, reaction and relation networks for metabolism, genetic information processing, environmental information processing, cellular processes, organismal systems, human diseases, and drug development. The analysis result was described in an illustration and the pathways involved in protein were described in a tabular format.

## Protein-protein interaction network analysis

Protein-protein interaction (PPI) analysis was conducted using the String Database (https://string-db.org/). The initial search focused on the target protein ERK2 within the protein query section, and the resulting interaction types and network were obtained in PNG format for further analysis and the pathways that were involved were retrieved in a table. The PPI network provides insight into which proteins might influence the inhibition of the target protein, offering a deeper understanding of the study.

# Results

## Ligand selection

Phytochemicals were chosen as primary ligands for the inhibition of the ERK protein due to their potential anti-cancer properties. Dr. Duke's database was utilized to extract ligands based on their anti-cancer, anti-carcinogenic, and cancer-preventive activities. Initially, 351 phytochemicals were identified. After removing duplicates, 340 unique compounds remained for further analysis.

## Prediction of active sites for ERK

The identification of active sites on the ERK protein was conducted using CASTP v3.0, a computational tool for locating and measuring pockets and voids on protein surfaces (http://sts.bioe.uic.edu/castp/calculation.html). This tool provided detailed information on the active sites, including the names and numbers of the residues involved. These active sites are critical regions on the protein where ligands, such as the selected phytochemicals, can bind. The CASTp has identified an active site of 489.308 $Å^2$ Area and a volume of 483.008 $Å^3$. A comparison was made between the active site features identified by the RCSB PDB structure (1TVO) and those identified by CASTp. The RCSB PDB provides the interacting residues information bound with the protein pocket, which indicates the active side of the protein. By comparing the active site features identified by 1TVO with those predicted by CASTp, we validated the accuracy and reliability of the site identification process. Both methods consistently identified key residues involved in ligand binding, confirming the suitability of the 1TVO structure for the docking studies. Differences observed in specific regions were analyzed to understand their impact on inhibitor binding and stability, providing deeper insights into potential binding mechanisms (Fig 1). The residue information is compiled into a comprehensive table (S1 Table). Additionally, the active site residues of ERK2 include hydrophobic (ILE, VAL, LEU, ALA, MET), polar (GLU, ASN, GLN, SER, THR), charged (LYS, ARG, ASP), and aromatic (TYR) amino acids, which are essential for ligand binding through hydrophobic interactions, hydrogen bonding, electrostatic interactions, and π-π stacking. These residues play a crucial role in the binding process, where the phytochemicals interact with the ERK protein, potentially inhibiting its activity.

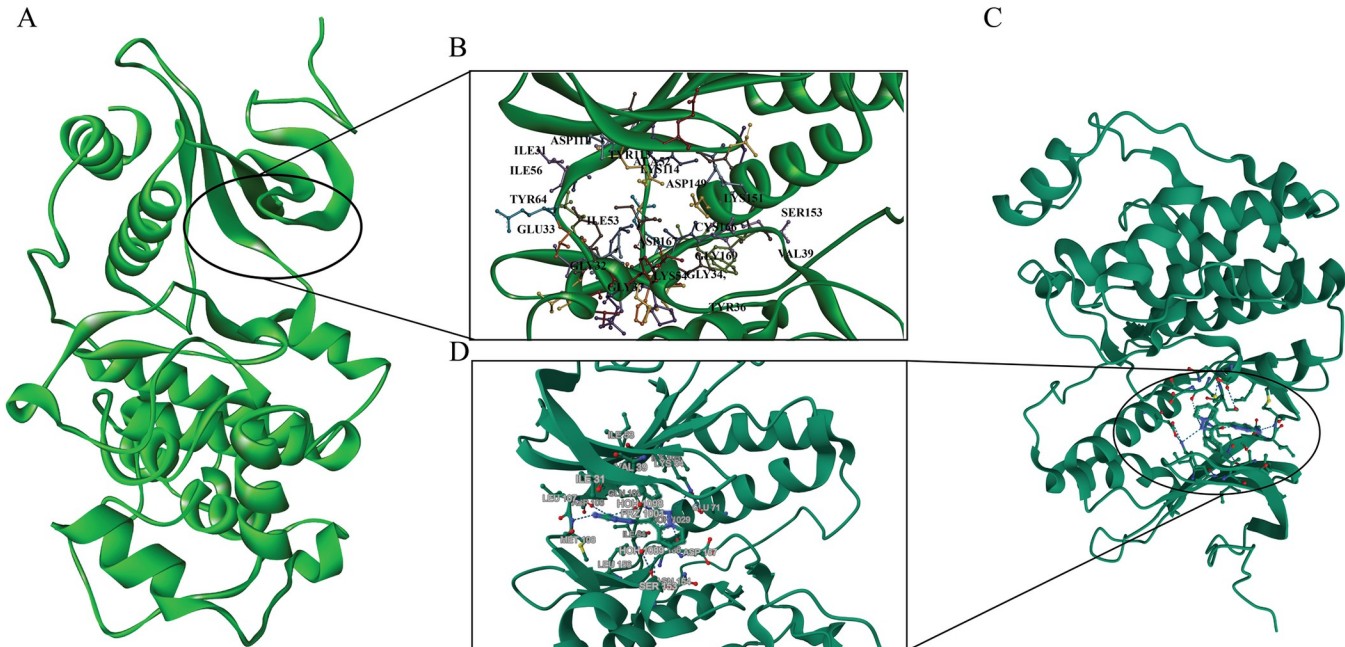

**Fig 1. The crystal structure of ERK protein's active site both from RCSB PDB and CASTp server.** The figure shows the structural analysis of a protein, highlighting the binding interactions at specific regions. The overall 3D structure of the protein, with the regions of interest marked as circled which have been identified from CASTp server (A). Zooms View and the active side residues have shown (B) with some key amino acid residues involved in binding interactions, such as val39 cys166, ser153, glu33, and tyr64, among others. The RCSB PDB has also checked for further validation (C). A close-up of this site illustrates the involvement of residues like ser153, val39, lys151, cys166, glu33 tyr64, and others (D). Some common residues indicate the same binding pocket involved in the prediction of the active side.

## ADMET profiling of sorting phytochemicals

ADMET profiling was performed for the sorted 340 phytochemicals. Among them, only 26 phytochemicals were filtered according to the threshold point (Table 2). Most of the phytochemicals did not satisfy the threshold point that was sought for further analysis (S1 File). The threshold points are poor absorption or permeation is expected to occur more frequently in the discovery setting when the number of H-bond donors exceeds 5, the number of H-bond acceptors reaches 10, the molecular weight (MWT) exceeds 500, and AMES toxicity and hepatotoxicity should be negative [36].

The Erk inhibitor exhibits hepatotoxicity, whereas the remaining compounds that satisfy all the associated criteria (Table 2) have been selected for subsequent processing. There is a correlation between median molecular weight and Heavy atom count. It was checked that the number of heavy atoms count < = 36 and corresponds to molecular weight < = 500 Dalton.

The findings suggest that the compound is suitable for further examination through docking and stability testing. The other parameters of ADMET profiling results such as molecular weight, heavy atoms, rotatable bonds, h-bond acceptors, h-bond donors, max. tolerated dose (human), skin sensitization, and minnow toxicity are given (S2 Table). All the AMDET results retrieved from the websites have been provided as S2 File.

## Cross-docking analysis with co-crystal ligand

Cross-docking studies are essential in validating the binding modes of ligands, ensuring that the computational docking methods accurately predict the interaction between ligands and their target proteins. The co-crystal ligand was docked into the binding site of a protein,

**Table 2. ADMET analysis result of selected compounds.**

| Sl no | Chemical | CID | XLOGP3 | Lipinski Rules 5 violation | BBB permeant | AMES toxicity | Hepato toxicity | GI absorption |
|---|---|---|---|---|---|---|---|---|
| 1 | Erk_inhibitor (control) | 135523966 | 4.1 | 0 | No | No | Yes | High |
| 2 | (+)-Catechin | 9064 | 0.36 | 0 | No | No | No | High |
| 3 | Apigenin | 5280443 | 3.02 | 0 | No | No | No | High |
| 4 | Aromadendrin | 122850 | 1.31 | 0 | No | No | No | High |
| 5 | Axillarin | 5281603 | 2.46 | 0 | No | No | No | High |
| 6 | Chrysoeriol | 5280666 | 3.1 | 0 | No | No | No | High |
| 7 | Cirsilineol | 162464 | 3.4 | 0 | No | No | No | High |
| 8 | Citrinin | 54680783 | 1.75 | 0 | No | No | No | High |
| 9 | CURCUMIN | 969516 | 3.2 | 0 | No | No | No | High |
| 10 | Diosmetin | 5281612 | 3.1 | 0 | No | No | No | High |
| 11 | Epicatechin | 72276 | 0.36 | 0 | No | No | No | High |
| 12 | Eriodictyol | 440735 | 2.02 | 0 | No | No | No | High |
| 13 | Eupatorin | 97214 | 3.4 | 0 | No | No | No | High |
| 14 | Galangin | 5281616 | 2.25 | 0 | No | No | No | High |
| 15 | GENISTEIN | 5280961 | 2.67 | 0 | No | No | No | High |
| 16 | Hesperetin | 72281 | 2.6 | 0 | No | No | No | High |
| 17 | Hispidulin | 5281628 | 2.99 | 0 | No | No | No | High |
| 18 | Isorhamnetin | 5281654 | 1.87 | 0 | No | No | No | High |
| 19 | Kaempferol | 5280863 | 1.9 | 0 | No | No | No | High |
| 20 | Luteolin | 5280445 | 2.53 | 0 | No | No | No | High |
| 21 | Melodorinol | 5388649 | 1.42 | 0 | No | No | No | High |
| 22 | Pelargonidin | 67249 | 3.1 | 0 | No | No | No | High |
| 23 | Quercetin | 5280343 | 1.54 | 0 | No | No | No | High |
| 24 | Rhamnetin | 5281691 | 1.87 | 0 | No | No | No | High |
| 25 | Rhein | 10168 | 2.23 | 0 | No | No | No | High |
| 26 | Scutellarein | 5281697 | 2.66 | 0 | No | No | No | High |
| 27 | Taxifolin | 439533 | 0.95 | 0 | No | No | No | High |

followed by a superimposition of the docked ligand with the original co-crystal structure. Both ligands occupy the same binding pocket, and key interaction residues (Fig 2). The overlapping region shows that both ligands maintain similar orientations, suggesting a conserved binding mode. Important residues such as Val39, Ala52, Asp106, and Cys166 are observed interacting with both ligands. A hydrogen bond, indicated by a green ellipse, is present, reinforcing the similarity in interaction patterns between the two ligands (Fig 2A and 2B). The RMSD value for the cross-docked pose 1 compared to the control (the co-crystal reference) is 0.578 Å, which is well below the 2 Å threshold, indicating high structural similarity.

## Molecular interaction at the active site

After filtering 26 phytochemicals by ADMET profiling, docking was performed in triplicates using two platforms AutoDock Vina-1.5.7 and PyRx (S3 Table). The control Ligands were also docked with the protein and recorded a binding score of -9.56±0.5 Kcal/mol (autodock vina) and -8.56±0.2 Kcal/mol (PyRx) for the positive control Pyrazolylpyrrole (CID135523966) which consider as the highest among the other inhibitors (Table 3). The other potent inhibitor has a score of -7.9±0.5 Kcal/mol (autodock vina) and -7.5±0.2 Kcal/mol (PyRx) for Ulixertinib (CID 11719003) and -9.2±0.1 Kcal/mol (autodock vina) and -8.8±0.1 Kcal/mol (PyRx) for Ravoxertinib (CID 71727581). Phytochemicals with higher binding affinity than the positive

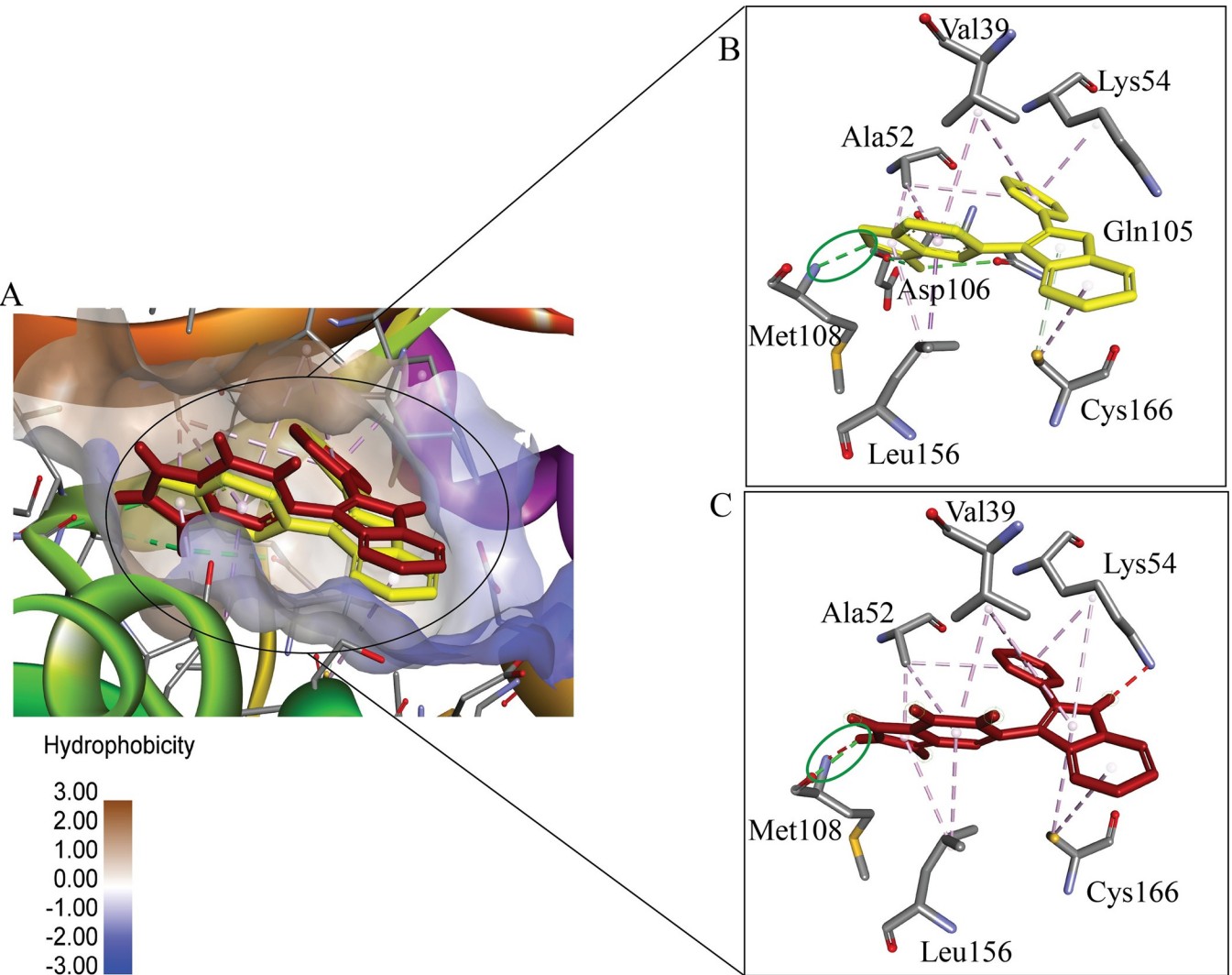

**Fig 2. Cross-docking analysis using the superimposed co-crystal and docked ligand in the binding pocket of the target protein.** (A) Displays the overall binding site with hydrophobicity mapping, while (B) and (C) provide zoomed views of the co-crystal and docked ligand interactions, respectively, highlighting key residues such as Val39, Ala52, Lys54, Asp106, and Cys166. Hydrogen bonds have been observed with Met108. The similar interactions and low RMSD values indicate the accuracy of the docking process in replicating the experimental binding mode.

control were considered potential inhibitors, and their interacting residues were analyzed further (Table 3).

Different binding residues including Val39, Ile31, Ala52, Glu71, Lys54, Ile103, Leu107, Cys166, Glu109, and Leu156 have shown interaction in multiple ligand complexes (Fig 3). Val39 showed hydrophobic interaction with the maximum of the complexes and the Ile103 which showed hydrogen bonds with most of the complexes. The interactions between the residues 2D also exhibited the ligand binding in a colored shape. The interacting complexes exhibit van der Waals interactions with residues that are not specified in the figure (Fig 3). The van der Waals interaction, being weaker, is responsible for the binding stability of the essential residues. Nevertheless, the hydrogen bond only forms with the oxygen atom of the ligand, as expected. Three unfavorable bonds have formed between complexes 3, 4, and 5, and the Lysine residues Lys114 (Fig 3C–3E). Complexes 3, 4, and 5 exhibited poor donor-donor interaction

**Table 3. The binding affinity of selected compounds (Kcal/mol) and the interacting residues.**

| Sl no | Chemical | CID | Docking Score (Autodock Vina) (Kcal/mol) | Docking Score (PyRx) (Kcal/mol) | Decoy Molecule Highest Binding Affinity (kcal/mol) | Hydrophobic Interaction | Bond Distance (Å) | Hydrogen /Electrostatic Bond | Bond Distance (Å) |
|---|---|---|---|---|---|---|---|---|---|
| 1 | Ulixertinib | 11719003 | -7.9±0.5 | -7.5±0.2 | N/A | Leu107 | 4.79404 | Met108 | 1.92094 |
| | | | | | | Val39 | 4.63365 | Glu109 | 2.13873 |
| | | | | | | Tyr36 | 4.05913 | Gly37 | 2.69181 |
| | | | | | | Ile31 | 4.72464 | Asp111 | 3.57627 |
| 2 | Ravoxertinib | 71727581 | -9.2±0.1 | -8.8±0.1 | N/A | Val39 | 4.50333 | Met108 | 2.81708 |
| | | | | | | Ala52 | 5.36445 | Gln105 | 2.66289 |
| | | | | | | Cys166 | 4.66778 | Gly67 | 3.18872 |
| | | | | | | Leu156 | 5.3124 | Asp167 | 3.47408 |
| 3 | Erk_Inhibitor (Control) | 135523966 | -9.56±0.5 | -8.56±0.2 | N/A | Tyr113 | 5.81217 | Cys166 | 3.56813 |
| | | | | | | Val39 | 4.70682 | Lys54 | 3.75414 |
| 4 | Quercetin (Complex 1) | 5280343 | -10.1±0 | -8.6±0.1 | -7.86±0.5 | Val39 | 4.21151 | Ile103 | 2.06415 |
| | | | | | | Ala52 | 5.24112 | | |
| | | | | | | Lys54 | 4.32546 | Cys166 | 3.76316 |
| | | | | | | Ile31 | 4.78653 | | |
| 5 | Apigenin (Complex 2) | 5280443 | -9.8±0.1 | -8.46±0.4 | -8.4±0 | Val39 | 4.67444 | Ile103 | 2.09527 |
| | | | | | | Ala52 | 4.87102 | | |
| | | | | | | Lys54 | 4.34842 | | |
| | | | | | | Ile31 | 4.84919 | | |
| 6 | Luteolin (Complex 3) | 5280445 | -10.1±0 | -8.46±0.1 | -7.5±0.1 | Val39 | 4.63344 | Ile103 | 2.12345 |
| | | | | | | Ala52 | 5.20102 | | |
| | | | | | | Lys54 | 4.35674 | | |
| | | | | | | Ile31 | 5.03245 | | |
| 7 | Chrysoeriol (Complex 4) | 5280666 | -10.2±0 | -8.36±0.2 | -8.33±0.5 | Ile31 | 4.05415 | Ile103 | 2.14996 |
| | | | | | | Leu107 | 5.02978 | | |
| | | | | | | Val39 | 4.65731 | | |
| | | | | | | Ala52 | 4.89949 | Glu109 | 3.29329 |
| | | | | | | Lys54 | 4.36711 | | |
| | | | | | | Ile31 | 5.02094 | | |
| 8 | Kaempferol (Complex 5) | 5280863 | -9.7±0 | -8.1±0.0 | -8.17±0.4 | Val39 | 4.67861 | Ile103 | 2.17298 |
| | | | | | | Ala52 | 4.96213 | Cys166 | 3.6809 |
| | | | | | | Lys54 | 4.35557 | | |
| | | | | | | Cys166 | 5.49784 | | |
| | | | | | | Ile31 | 4.87128 | | |
| 9 | Genistein (Complex 6) | 5280961 | -9.8±0.1 | -8.66±0.1 | -8.4±0 | Ile31 | 3.89119 | Glu71 | 2.74503 |
| | | | | | | Ala52 | 4.59047 | | |
| | | | | | | Leu156 | 5.45418 | | |
| | | | | | | Val39 | 4.98374 | Lys54 | 2.43912 |
| | | | | | | Lys54 | 4.76887 | | |
| | | | | | | Cys166 | 5.28675 | | |
| 10 | Hispidulin (Complex 7) | 5281628 | -9.86±0.15 | -8.4±0.0 | -7.86±0.01 | Val39 | 4.62924 | Glu71 | 2.4999 |
| | | | | | | Cys166 | 5.30716 | Asp167 | 2.26345 |
| | | | | | | Ala52 | 4.24949 | Met108 | 2.25224 |
| | | | | | | Leu156 | 4.64733 | Lys54(ES) | 4.32009 |

(*Continued*)

**Table 3.** (Continued)

| Sl no | Chemical | CID | Docking Score (Autodock Vina) (Kcal/mol) | Docking Score (PyRx) (Kcal/mol) | Decoy Molecule Highest Binding Affinity (kcal/mol) | Hydrophobic Interaction | Bond Distance (Å) | Hydrogen /Electrostatic Bond | Bond Distance (Å) |
|---|---|---|---|---|---|---|---|---|---|
| 11 | Isorhamnetin (Complex 8) | 5281654 | -9.76±0.025 | -8.2±0.0 | -8.06±0.01 | Val39 | 3.97917 | Glu71 | 2.395 |
| | | | | | | Ala52 | 5.33661 | Ile103 | 1.95507 |
| | | | | | | Lys54 | 4.68671 | Asp111 | 2.71769 |
| | | | | | | Val39 | 4.46223 | Ser153 | 3.42483 |
| 12 | Rhamnetin (Complex 9) | 5281691 | -10.0±0 | -8.46±0.1 | -8.9±0 | Leu107 | 4.96767 | Asp111 Ile103 | 2.00107 2.31747 |
| | | | | | | Ile31 | 4.6251 | | |
| | | | | | | Val39 | 5.34652 | | |
| | | | | | | Ala52 | 5.24105 | Glu109 Lys54 | 3.14519 3.49135 |
| | | | | | | Leu156 | 5.05816 | | |
| | | | | | | Lys54 | 4.32332 | | |

with Lys114. Control (A) and complex 7 (H) both exhibit pi-cation interactions with Lys151 and Lys52 residues. While some van der Waals interactions are also present, they are depicted in a lighter green tone and do not involve bonding interactions. This is because they produce weaker bonds compared to other types of bonding.

## Decoy screening of phytochemicals

Decoy screening was conducted as a negative control by docking 50 decoy molecules for each compound in a triplicate manner (S3 File). The decoy molecules, which share similar physical properties with the active compounds but differ structurally, serve to assess the specificity of the phytochemicals as inhibitors. Phytochemicals such as quercetin (Complex 1), luteolin (Complex 3), and hispidulin (Complex 7) demonstrated significantly higher binding affinities (-10.1 kcal/mol and 9.86 kcal/mol) compared to their decoys (-7.86 kcal/mol, -7.7 kcal/mol and -7.86 kcal/mol, respectively), indicating strong specificity (Table 3). The decoy screening identified rhamnetin (Complex 9) as the compound where the decoys failed. Specifically, rhamnetin showed a binding affinity of -8.46 kcal/mol, but its decoy exhibited a relatively high binding affinity of -8.9 kcal/mol. Overall, phytochemicals with binding affinities substantially lower than their decoys are considered more specific inhibitors, while those with comparable or lower specificity may involve non-specific binding, as seen in the case of rhamnetin. This failure indicates that rhamnetin may have non-specific interactions, reducing its reliability as a specific inhibitor.

## Molecular dynamics simulation analysis

A simulation lasting 200 nanoseconds was conducted to assess the stability of the binding. Typically, simulations of up to 100 nanoseconds are performed in most simulation studies. However, it is feasible to demonstrate the stability or instability of binding after 100 nanoseconds. The simulation results were evaluated using RMSD, RMSF, SASA, and radius of gyration analysis. The binding grooves of the compounds under investigation were compared and found to have a significant level of similarity in their spatial layouts. Only the complexes exhibiting better RMSD, RMSF, SASA, and gyration metrics during MD simulation were shown in the binding grooves. These stable complexes demonstrated deep binding grooves with shared common residues for interaction (Fig 4). Furthermore, the residues that were involved in interactions showed remarkable similarity across all the compounds. The presence of congruency in the binding grooves and interacting residues indicates a consistent way of binding,

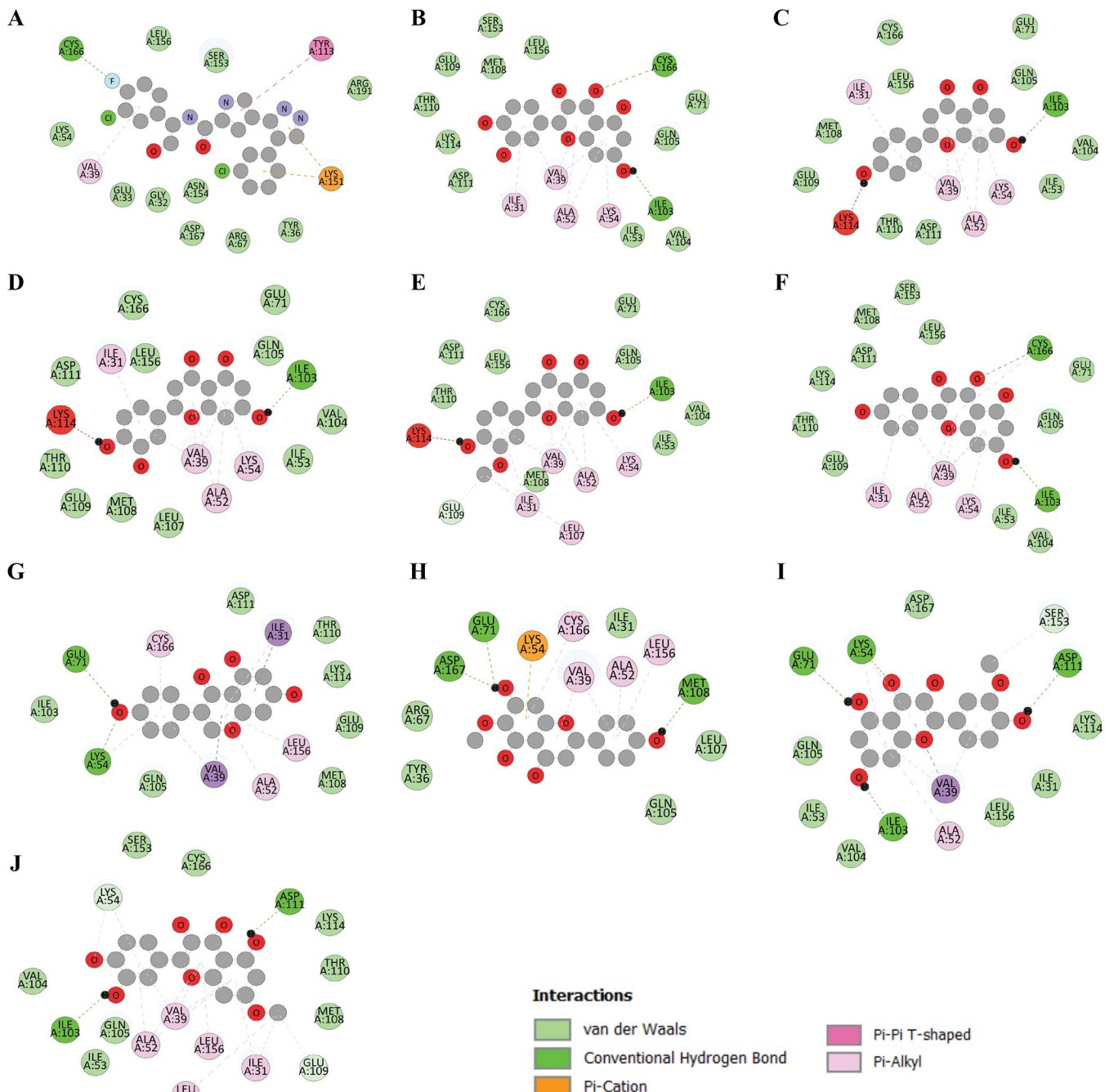

**Fig 3. Two-dimensional interactions with the selected compounds with ERK protein.** The control (A) along with all nine other complexes (B-J) with higher affinity were analyzed for their interacting residues. The color code provided below the figure indicates different types of chemical interactions such as van der Waals, conventional hydrogen bonds, Pi-cation, Pi-Pi t-shaped, and Pi-alkyl bonds.

which strengthens the probability of a common molecular process or target interaction. The pocket region of the complex (Fig 4) depicts all the ligands that were bound in the same pocket and the residues that mimicked those 3 complexes were val39, ala52, ile103, and lys54. Among these bindings, complex 7 hispidulin (Fig 4D) has an electrostatic bond with lys54. The overall binding interactions are quite good.

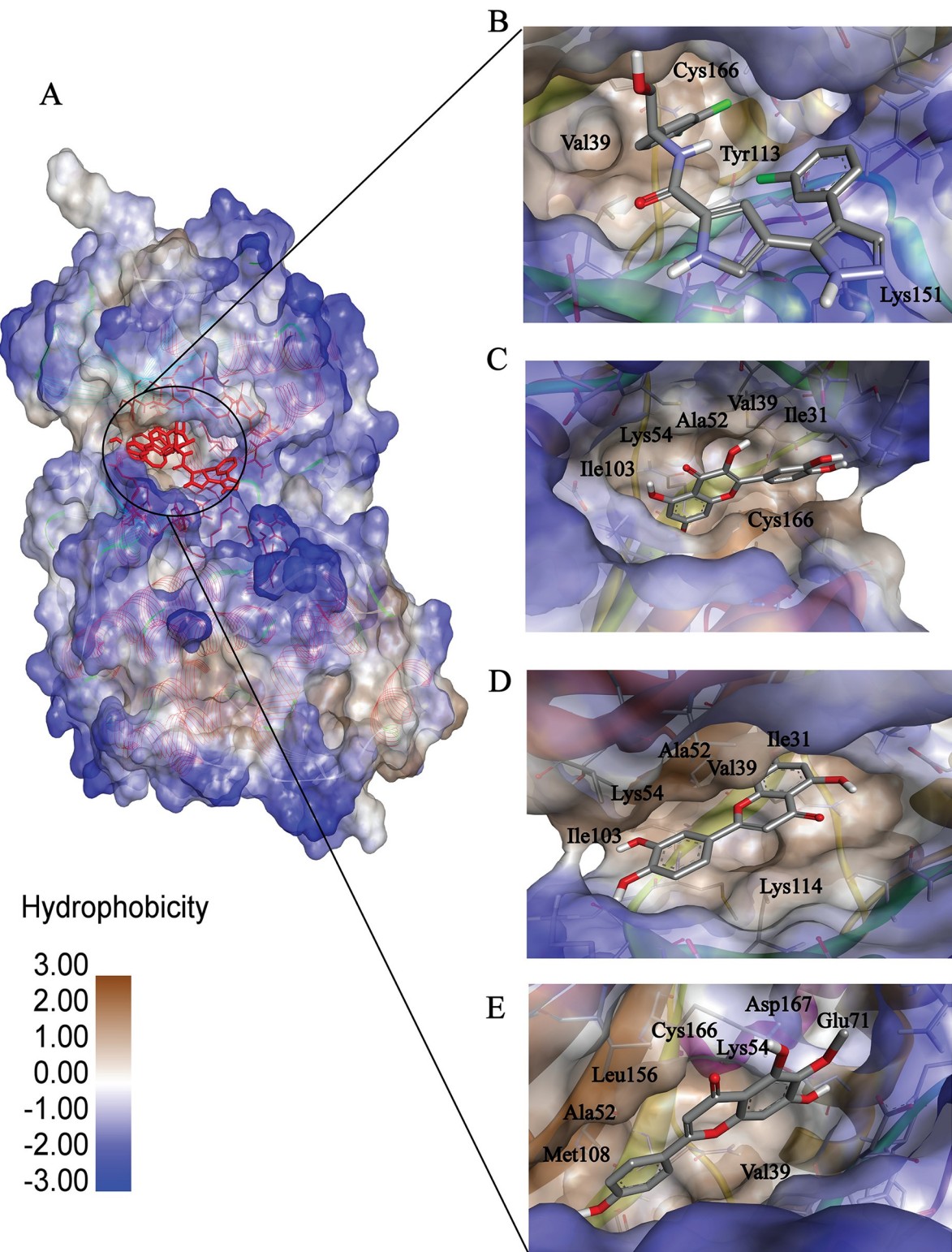

**Fig 4. A visual representation of the binding pocket and ligand interaction.** (A) The 3d Structure of protein-ligand complex and protein hydrophobicity mapping. Close view of Control (B), Complex_3 (C), Complex_7 (D) and Complex_8 (E). The protein pocket region is slightly bluish which indicates partially hydrophilic. All the ligands bind to the same side of the protein. Additionally, the binding affinity seems to be influenced by the presence of hydrogen bond donors and acceptors in the ligands and the polarity of the ligand influences the binding.

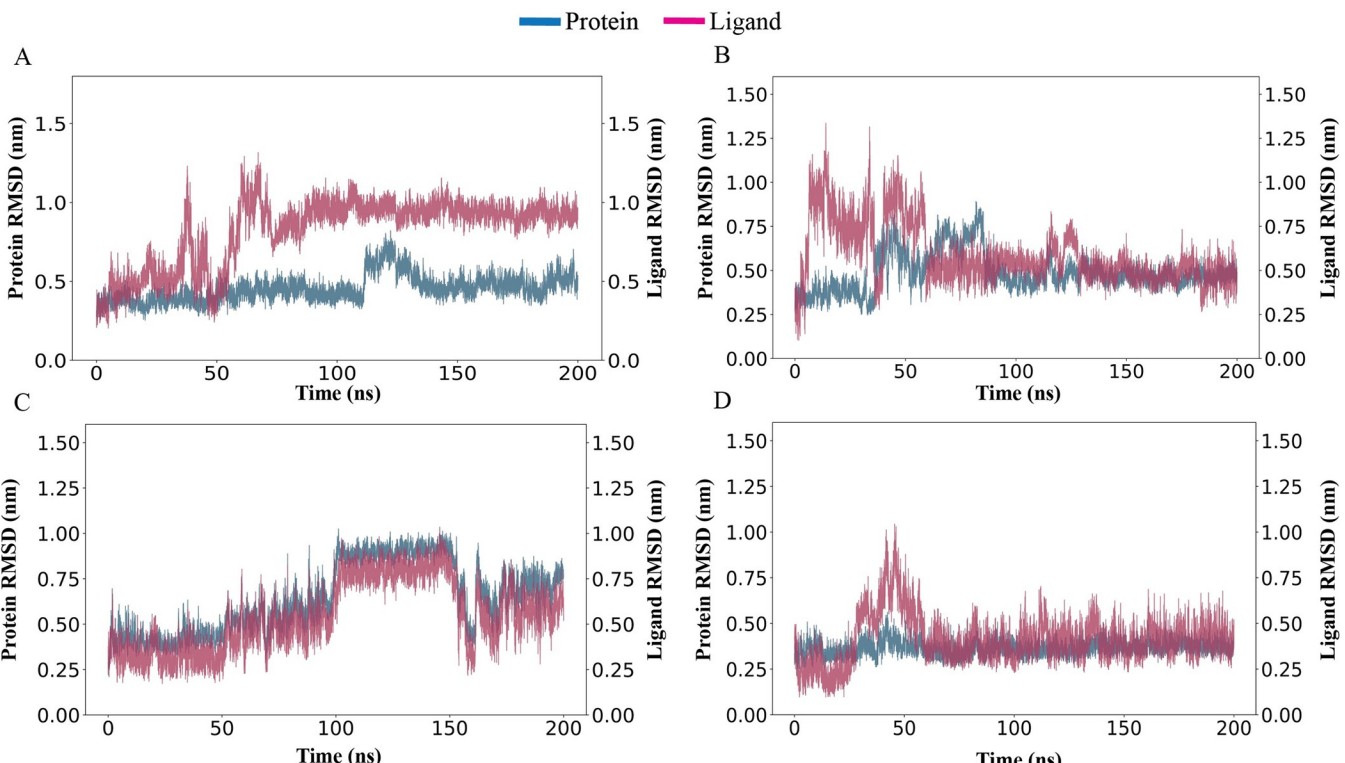

**Fig 5. A 200-nanosecond simulation is conducted to measure the root mean square deviation (RMSD).** Complexes 3, 7, and 8 along with the control were subjected to a 200-nanosecond molecular dynamics simulation using the Gromacs software. The root means square deviation (RMSD) between the ligand and protein exhibits temporal constancy, thereby ensuring stability. A) RMSD of Control, B) RMSD of Complex 3, C) RMSD of Complex 7, and D) RMSD of Complex 8.

The RMSD of Protein-ligand complexes have shown the Protein RMSD fit with ligand RMSD over a 200ns time scale. RMSD, which is the ligand insect in the protein RMSD line, is considered a good stability benchmark. Control which is considered a potent inhibitor has some deviation (Fig 5A) during the time whereas Complex_3, Complex_7, and Complex_8 show better binding stability (Fig 5B–5D). The control complex (Fig 4A) exhibits notable deviations, with protein RMSD ranging from 0.2 nm to 1.0 nm and ligand RMSD from 0.4 nm to 1.2 nm, indicating less stable binding. Complex 3 (Fig 5B) initially shows deviations with protein RMSD up to 1.0 nm and ligand RMSD up to 1.2 nm but stabilizes after 100 ns to 0.4–0.6 nm for both. Complexes 7 (Fig 5C) and 8 (Fig 5D) demonstrate consistent stability throughout the simulation, with protein RMSD around 0.4–0.6 nm and ligand RMSD around 0.4–0.8 nm. These results suggest that Complexes 7 and 8 have better binding stability compared to the control and Complex 3, which stabilizes only after an initial period of deviation. The overall binding interaction for all complexes appears stable, but Complexes 7 and 8 show the most consistent stability, indicating strong protein-ligand interactions. Further investigations are required to fully understand the binding characteristics and fluctuations in these complexes. The other six complexes showed poor stability based on the simulation results (S1 Fig).

The complex Root Mean Square Fluctuation (RMSF) is a valuable tool for quantifying localized variations along the protein chain. Peaks on the plots represent regions of the protein that exhibit the highest degree of fluctuation throughout the simulation. It is commonly observed that the tails, specifically the N- and C-terminal, exhibit greater fluctuations compared to other regions of the protein. Secondary structure elements, such as alpha helices and beta strands,

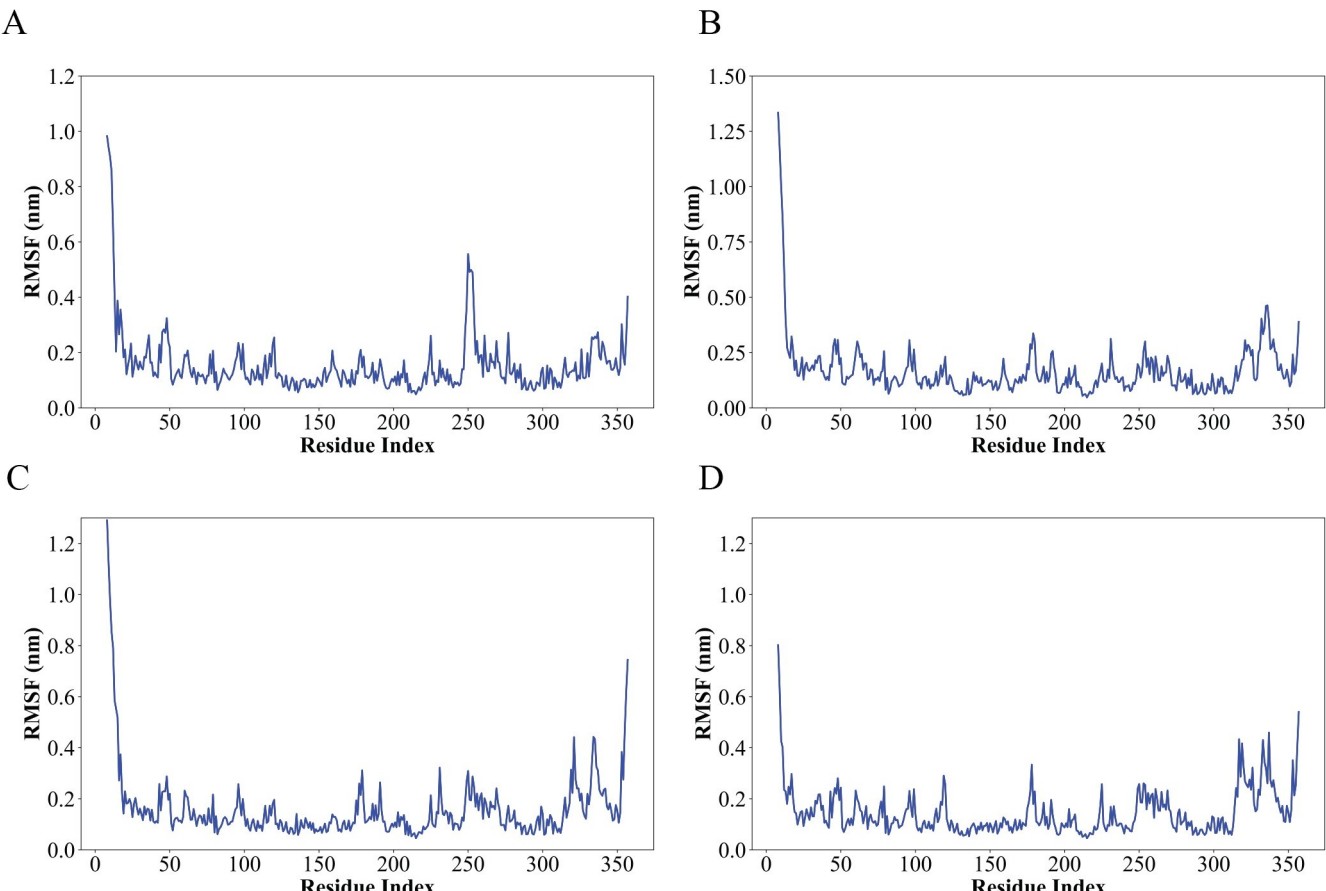

**Fig 6. The root means square fluctuation (RMSF) of all the simulation complexes over a 200-nanosecond simulation.** A- Root Mean Square Fluctuation (RMSF) of Control, B- RMSF of Complex 3, C- RMSF of Complex 7, and D- RMSF of Complex 8. The interpretation of the results is justified. The fluctuation primarily arises when the ligand interacts with the protein residues. Complex 3 exhibits three significant fluctuations. Complex 7 and Complex 8 exhibit significant temporal fluctuations. The overall comparison reveals all the fluctuations, although they do not exceed 0.6 nm.

typically exhibit greater rigidity compared to the unstructured regions of the protein. As a result, they undergo less fluctuation than the loop regions (Fig 6). The root means square fluctuations (RMSF) of the residues in complexes 3, 7, and 8 have revealed significant changes between residues 340 and 350. However, the overall fluctuations of 250 residues do not surpass 0.6nm, and there are no infinite fluctuations in the interaction with the ligand (Fig 6). Nevertheless, complexes 7 and 8 demonstrate persistent stability, suggesting that the interaction between the protein and ligand remains intact throughout the entire duration. Complex 3 exhibits a deviation up to 100ns, indicating inferior stability compared to the other 2 complexes, but after 100 ns the binding of protein and ligand goes in the direction of stability. The other six complexes exhibit several significant fluctuations based on the simulation RMSF analysis (S2 Fig).

The SASA analysis over the 200 ns simulation reveals distinct differences in solvent accessibility between the unbound and ligand-bound states of the protein. The unbound protein (Only protein) consistently exhibits lower SASA values, averaging around 140–150 nm$^2$, indicating a more compact and less solvent-exposed conformation. In contrast, the ligand-bound states (Complex_3, Complex_7, Complex_8, and Control) show higher SASA values, averaging around 170–180 nm$^2$, suggesting that ligand binding induces conformational changes that increase the surface area exposed to the solvent. These observations indicate that ligand

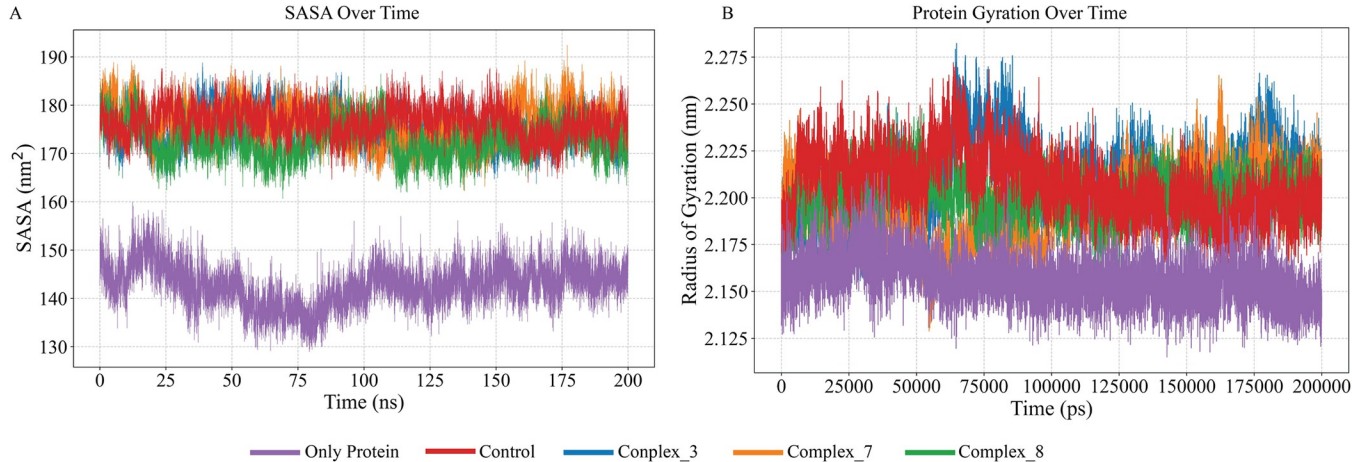

**Fig 7. Solvent Accessible Surface Area (SASA) and radius of gyration (Rg) over time for various protein-ligand complexes and control.** The graph shows the SASA values (in $nm^2$) (6A) and the radius of gyration (in nm) (6B) plotted over a 200 ns molecular dynamics simulation for different systems, Complex_3 (blue), Complex_7 (orange), Complex_8 (green), Control (red), and Only Protein (purple). The data illustrates the dynamic changes in the solvent exposure of the protein in the presence of different ligands and the unbound state. The complexes with ligands (Complex_3, Complex_7, Complex_8, and Control) exhibit higher SASA values, indicating more solvent-exposed surfaces compared to the free protein, which maintains a consistently lower SASA throughout the simulation. The Rg, measured in nanometers (nm), is an indicator of the protein's compactness and structural stability over time. A smaller radius of gyration indicates a protein structure that is more tightly packed, while a larger radius of gyration denotes a structure that is more spread out or unfolded. A smaller radius of gyration signifies a denser protein structure, while a larger radius of gyration indicates a more elongated or unfolded structure.

binding causes the protein to adopt a more open or extended conformation, which results in increased solvent exposure. This supports the hypothesis that ligand interaction is associated with structural reorganization, enhancing the solvent accessibility of the protein surface (Fig 7A). The SASA information of the remaining six complexes showed poor stability during RMSD analysis (S3 Fig).

The radius of gyration is calculated as the distance between the center of mass of all the atoms in the protein and its extremities over a specific time interval. Proteins with a smaller

**Table 4. The MAPK1/ERK2 protein involvement in the different pathway regulation.**

| Sl no | KEGG Pathway ID | Pathway description | Observed gene count | False discovery rate | Matching proteins |
|---|---|---|---|---|---|
| 1 | hsa05223 | Non-small cell lung cancer | 5 | 2.31E-08 | MAPK1, MAP2K2, STAT3, TP53, MAP2K1 |
| 2 | hsa05167 | Kaposi sarcoma-associated herpesvirus infection | 6 | 2.43E-08 | MAPK1, MAP2K2, STAT3, TP53, MAP2K1, JUN |
| 3 | hsa05210 | Colorectal cancer | 5 | 3.78E-08 | MAPK1, MAP2K2, TP53, MAP2K1, JUN |
| 4 | hsa05235 | PD-L1 expression and PD-1 checkpoint pathway in cancer | 5 | 4.31E-08 | MAPK1, MAP2K2, STAT3, MAP2K1, JUN |
| 5 | hsa05216 | Thyroid cancer | 4 | 1.67E-07 | MAPK1, MAP2K2, TP53, MAP2K1 |
| 6 | hsa05219 | Bladder cancer | 4 | 1.82E-07 | MAPK1, MAP2K2, TP53, MAP2K1 |
| 7 | hsa05224 | Breast cancer | 5 | 2.64E-07 | MAPK1, MAP2K2, TP53, MAP2K1, JUN |
| 8 | hsa05206 | MicroRNAs in cancer | 5 | 3.51E-07 | MAPK1, MAP2K2, STAT3, TP53, MAP2K1 |
| 9 | hsa05213 | Endometrial cancer | 4 | 5.69E-07 | MAPK1, MAP2K2, TP53, MAP2K1 |
| 10 | hsa05205 | Proteoglycans in cancer | 5 | 7.76E-07 | MAPK1, MAP2K2, STAT3, TP53, MAP2K1 |
| 11 | hsa05211 | Renal cell carcinoma | 4 | 7.76E-07 | MAPK1, MAP2K2, MAP2K1, JUN |
| 12 | hsa05221 | Acute myeloid leukemia | 4 | 8.01E-07 | MAPK1, MAP2K2, STAT3, MAP2K1 |

radius of gyration exhibit higher packing density and greater compactness, measured by the ratio of their accessible surface area to that of a perfect sphere with the same volume. The data result found after Rg analysis showed that the radius of gyration for the only protein condition remains consistently lower, with an average Rg of approximately 2.15 nm, suggesting a more compact and stable structure. In contrast, the complexes exhibit higher Rg values, with Complex_3, Complex_7, Complex_8, and the Control having average Rg values around 2.22 nm, 2.20 nm, 2.21 nm, and 2.23 nm, respectively (Fig 7B). These higher and more variable Rg values indicate that the protein in these complexes adopts a less compact and more dynamic structure. The findings suggest that the presence of different ligands or interactions in the complexes impacts the protein's structural stability, potentially affecting its biological function. In conclusion, the increased Rg values in the complexes highlight the significant role of ligand binding in modulating protein conformation and stability (Fig 7B). The Gyration information of the remaining six complexes showed poor stability during RMSD analysis (S4 Fig).

Proteins with moderate compactness, indicated by their radius of gyration, showed increased gyration values upon ligand binding due to structural disruption. The Solvent Accessible Surface Area (SASA) analysis revealed that moderate gyration and higher molecular surface area provided multiple binding sites, while moderate SASA indicated structural stability. Lowering gyration (Fig 7B) and SASA (Fig 7A) is necessary for stronger binding affinity.

## Metabolic pathway analysis

The current research involves the examination of metabolic pathways, specifically focusing on the RAS signaling pathway. During this analysis, an essential enzyme called MAPK1/3 or ERK1/2, which is activated by a group of enzymes known as RTK (Receptor tyrosine kinase) is stimulated by certain extracellular signaling molecules [37]. The signal triggers the transformation of RAS (rat sarcoma) protein from RAS-GDP to RAS-GTP, which is its active form. This, in turn, activates cRAF (cytosolic Rapidly Accelerated Fibrosarcoma kinase), a key component of the mitogen-activated protein kinase (MAPK) pathway. c-Raf subsequently transmits the signal to MEK (Mitogen-activated protein kinase kinase) by phosphorylation of the protein [38]. Then the signal is transmitted to the ERK1/2 (extracellular signal-regulated kinase) protein, which serves as the primary enzyme under investigation in this work. The ERK proteins (1/2) play a direct role in gene regulation. Cell survival is heavily reliant on this signaling system, specifically the upregulation of the ERK proteins which is involved in gene regulation [39]. Targeting the overexpression of this protein could be a promising approach for inhibiting and treating uncontrolled cell proliferation. Despite the crucial role of the ERK proteins in gene regulation and cell survival, our analysis suggests that targeting its overexpression could be an effective strategy for inhibiting and treating uncontrolled cell proliferation [40]. By understanding the intricacies of the RAS signaling pathway and the pivotal role of ERK within it, we can develop more precise therapeutic interventions aimed at mitigating diseases characterized by excessive cell growth, such as cancer.

Depending on the cellular context, the MAPK/ERK cascade mediates diverse biological functions such as cell growth, adhesion, survival, and differentiation through the regulation of transcription, translation, and cytoskeletal rearrangements (Table 4). The Protein-Protein Networking Depicts the insight of the MAPK/ERK cascade. The focus is MAPK1/ERK2 isoforms, the main pathway here can observe different types of cancer which is an important ask for these studies.

## PPI networking analysis

The protein-protein interaction (PPI) network analysis, performed using the STRING database, revealed significant interactions within the MAPK/ERK cascade pathway. MAPK1 (also

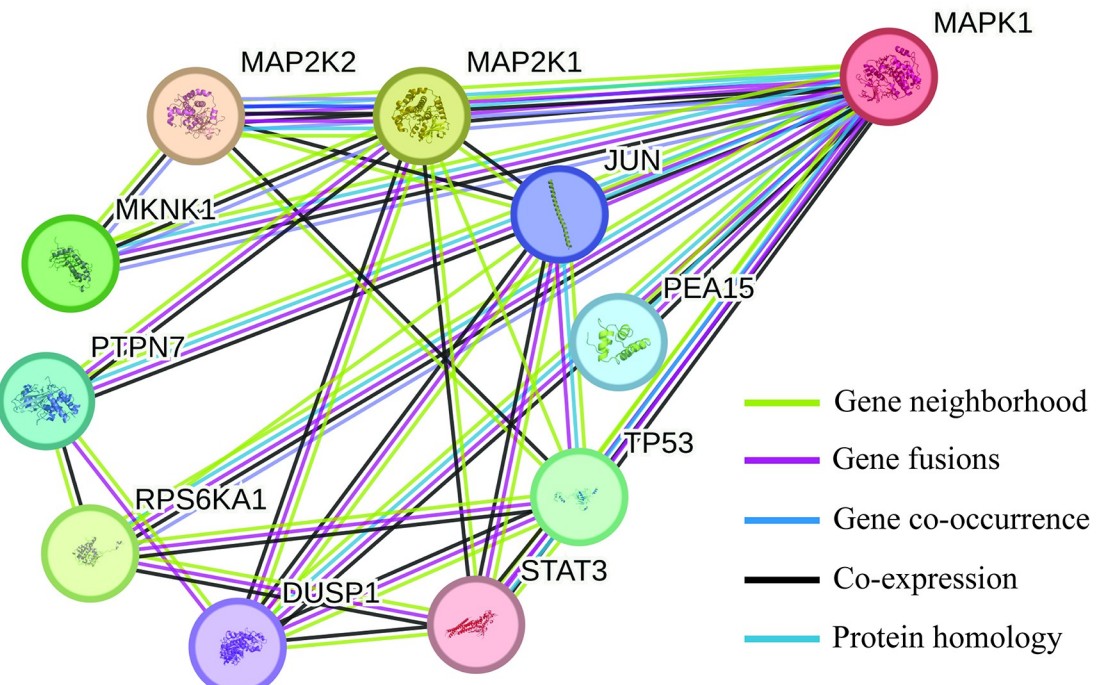

**Fig 8. A PPI networking of MAPK1/ERK2 isoform of ERK protein.** Protein MAPK1 is involved in different protein regulations and different disease pathways. The main gene protein interactions are covered as gene neighborhoods colored in green lines, gene fusions colored in red lines, and gene co-occurrence in blue lines. Protein homology is colored with cyan, and the co-expression is colored with black lines.

known as ERK2) is a critical protein in this pathway, interacting with various other proteins that play essential roles in cellular processes. The PPI network highlights several key interactions and their implications, particularly focusing on TP53, a tumor suppressor protein [41]. The Protein MAPK1/ERK2 interacts with TP53 acts as a tumor suppressor and is involved in cell cycle regulation by inducing growth arrest or apoptosis, STAT3 which is involved in cell growth and apoptosis [42–44]. The JUN-A component of the transcription factor AP-1 is involved in regulating gene expression in response to a variety of stimuli. RPS6KA1- participates in signaling pathways related to cell growth and survival. DUSP1 acts as a negative regulator of MAPK activity, providing feedback inhibition. PEA15 regulates cell proliferation and apoptosis. Inhibiting MAPK1 may reduce the expression and activity of proteins that are co-expressed with it, such as JUN and STAT3 (Fig 8).

This can lead to diminished cellular responses to external stimuli and potentially reduce cell proliferation and survival signals. Co-activators like TP53 and RPS6KA1 could experience altered functionality. TP53 may become less effective in inducing apoptosis or growth arrest in response to cellular stress, which could impact tumor suppression activities. RPS6KA1's role in promoting cell growth and survival may be compromised, potentially leading to reduced cancer cell proliferation. Some Proteins involved in feedback regulation, such as DUSP1, might show altered activity. The inhibition of MAPK1 could lead to changes in the feedback loops that control MAPK pathway activity, potentially resulting in increased or decreased MAPK signaling.

## Discussion

The present study leverages molecular docking and simulation techniques to identify and analyze the inhibitory potential of selected phytochemicals against the MAPK/ERK pathway, a

critical component in the RAS signaling cascade [41]. The MAP kinase superfamily comprises extracellular signal-regulated kinases 1 and 2 (ERK1/2), which play a crucial role in controlling cell proliferation and survival [44]. Specifically, ERK proteins have been reported to enhance cell viability by inhibiting the function of caspase 9 [45, 46].

The ERK/MAPK signaling system is regulated by several stimulating stimuli, including cytokines, viruses, G-protein-coupled receptor ligands, and oncogenes [47]. The ERK/ MAPK signaling pathway can be activated through the following mechanisms: The following are the four types of activation: i) $Ca^{2+}$ activation; ii) receptor tyrosine kinase Ras activation; iii) PKC-mediated activation; and iv) G protein-coupled receptor activation [48, 49]. Additionally, this route is closely associated with tumor formation. Increased levels of ERK [50] expression have been observed in different types of human tumors, including ovarian, colon, breast, and lung cancer [51, 52]. Normal ovarian surface epithelium and benign cystadenomas express more MKP-1 than invasive carcinomas, low malignancy potential tumors, and borderline tumors [53]. MKP-1 expression in tumor tissues of advanced-stage (III/IV) patients was significantly lower than that of early-stage (I/II) patients. p-ERK1/2 levels were much higher in normal ovarian tissues, benign tumors, and borderline tumors. Stage III/IV patients had significantly higher p-ERK1/2 expression than stage I/II patients. Immunohistochemistry and western blotting showed an inverse relationship between MKP-1 and p-ERK1/2 in ovarian cancer tissue. Tumors and cancer cells may benefit from this protein study [54–56].

From an initial study of a dataset of 351 phytochemicals identified through Dr. Duke's Phytochemical and Ethnobotanical Databases, after removing the redundancy 340 phytochemicals are sorted to conduct the study. Ligands that did not meet the following parameters were excluded Molecular Weight (120–500 g/mol), Rotatable Bond Count (1–10), Heavy Atom Count (12–30), H-Bond Donor Count (0–4), H-Bond Acceptor Count (0–10), GI Absorption (High), Polar Area (4.9–104 Å$^2$), Complexity (144–494), XLOGP (1–5), and compliance with Rule of 5 with no violations. Additionally, the toxicity and sensitivity parameters have addressed, confirming that excluded compounds did not pose risks for AMES toxicity, hepatotoxicity, or skin sensitivity. Out of 340, 26 compounds were selected based on stringent ADMET profiling criteria. These criteria ensured favorable pharmacokinetic properties and low toxicity. All selected compounds exhibited high gastrointestinal absorption, no violations of Lipinski's Rule of Five, and showed no hepatotoxicity or AMES toxicity, indicating good oral bioavailability and safety as potential therapeutic agents. The docking results revealed that these compounds interacted with key residues within the ERK2 binding pocket, indicating their ability to modulate the activity of this kinase. Notably, the high binding affinities observed for these phytochemicals suggest that they could effectively inhibit the MAPK/ERK pathway, thereby potentially preventing the proliferation of RAS-driven tumors [57, 58]. The low RMSD values (<2 Å) between the docked and co-crystal ligands indicate that the docking protocol effectively reproduces the binding pose of the ligand as observed in the experimental structure [59].

Quercetin (CID 5280343), apigenin (CID 5280443), luteolin (CID 5280445), genistein (5280961), hispidulin (CID 5281628), isorhamnetin (5281654), chrysoeriol (CID 5280666) and rhamnetin (CID 5281691) emerged as the top candidates, demonstrating binding affinities comparable to or better than the positive control N-[1-(3-Chloro-4-fluorophenyl)-2-hydroxyethyl]-3-[4-(3-chlorophenyl)-1,2-dihydro-3H-pyrazol-3-ylidene]-3H-pyrrole-5-carboxamide (Pyrazolylpyrrole) (CID 135523966). However, the decoy screening revealed that while most phytochemicals exhibited higher specificity compared to their decoys, rhamnetin (CID 5281691) showed a concerning result (Table 3). The decoy for rhamnetin had a relatively high binding affinity of -8.9 kcal/mol, lower than the actual compound's -8.46 kcal/mol, indicating

potential non-specific interactions. The binding interactions involved key residues within the ERK2 active site, including Val39, Ile103, Ala52, and Lys54. These interactions suggest that the phytochemicals can effectively inhibit ERK2 activity by stabilizing the inactive conformation of the protein.

The stability of these interactions, as demonstrated by simulation studies with a 200ns time scale, further supports the therapeutic promise of these compounds (Figs 5–7). These findings align with previous research indicating the therapeutic potential of natural compounds in targeting protein kinases involved in cancer pathways. Comparing the docking and simulation results with the natural ligand and positive control, the phytochemicals exhibited competitive binding affinities and stable interactions. These findings indicate that phytochemicals, particularly flavonoids such as luteolin, hispidulin, and isorhamnetin exhibit significant binding affinity to ERK2 in RMSD RMSF SASA and gyration studies (Figs 5–7). The predicted ligands luteolin, hispidulin, and isorhamnetin demonstrate strong potential as ERK1/2 inhibitors due to their multiple hydroxyl and methoxy groups, which enable versatile hydrogen bonding interactions. Luteolin's hydroxyl groups, hispidulin's additional methoxy group, and isorhamnetin's unique 3'-methoxy group contribute to their effective binding dynamics. Compared to known inhibitors like ulixertinib and ravoxertinib, which feature nitrogen heterocycles and halogens for strong binding. The comparison also revealed that all ligands share crucial structural motifs, such as aromatic rings, which are essential for interactions within the ATP-binding pocket of ERK1/2. However, differences were observed in the overall molecular flexibility and side chain composition, where the known inhibitors exhibit greater conformational adaptability due to flexible aliphatic chains (Fig 9). These findings suggest that the phytochemical ligands retain the necessary features for effective binding, they may engage the target protein

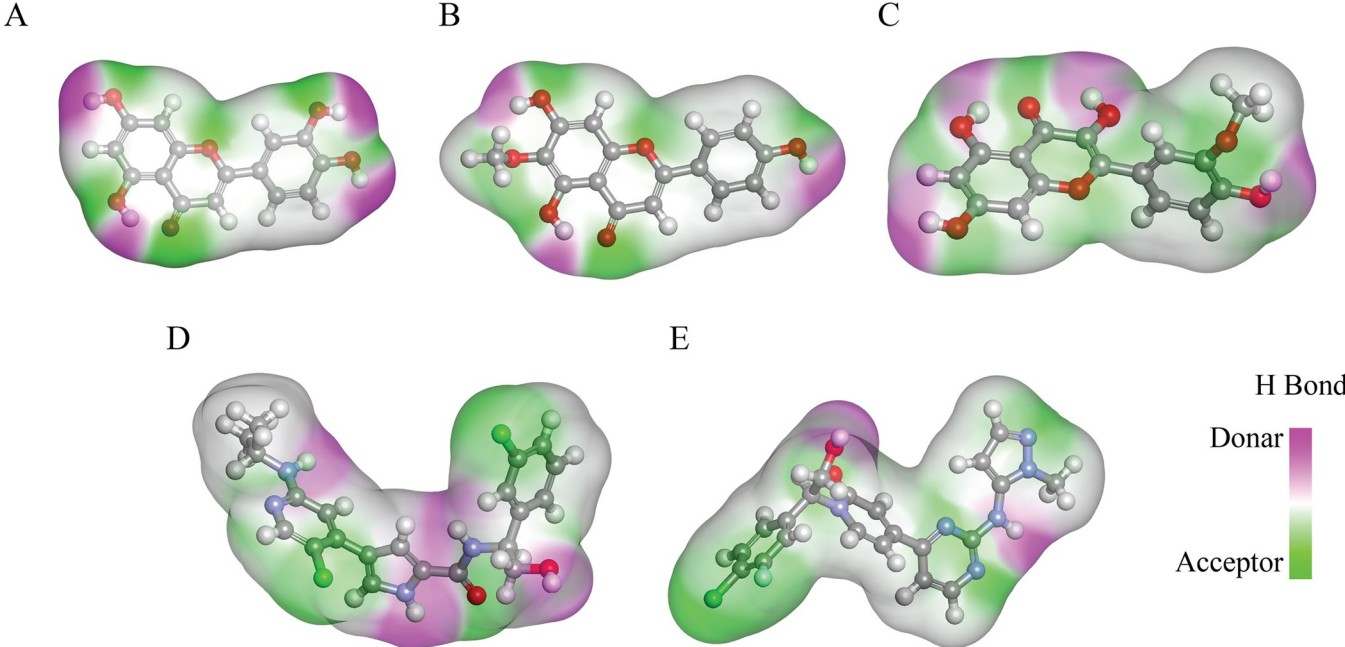

**Fig 9. Molecular structures and comparison between predicted ligand and known inhibitors.** (A) Luteolin (CID 5280445), (B) Hispidulin (CID 5281628), and (C) Isorhamnetin (CID 5281654) represent our predicted ligands. (D) Ulixertinib (CID 11719003) and (E) Ravoxertinib (CID 71727581) are known ERK1/2 inhibitors. The colors in the molecular structures denote different atoms: red for oxygen, blue for nitrogen, white for hydrogen, green for chlorine, and light green for fluorine. The surrounding contours highlight hydrogen bond donors and acceptors, where purple indicates donors and green indicates acceptors.

in a manner distinct from the known inhibitors, potentially leading to different inhibitory effects.

It is crucial to note that while in silico studies provide valuable insights, experimental validation through in vitro and in vivo studies is essential to confirm the efficacy and safety of these compounds in biological systems [60]. The RAS-ERK signaling pathway, emphasizes key proteins, mutations, and potential points of inhibition. The pathway begins at the cell surface with Receptor Tyrosine Kinase (RTK), which, upon activation by extracellular signals, triggers the transformation of RAS from its inactive form (RAS-GDP) to its active form (RAS-GTP) [61]. Activated RAS then stimulates CRAF, which phosphorylates MEK. Mutations in BRAF (such as V600E and L597Q) can also lead to constitutive activation of MEK, independent of upstream signals, and these mutations can be targeted by BRAF inhibitors (BRAFi) [62, 63].

MEK, once activated, phosphorylates ERKs. Phosphorylated ERKs are translocated to the nucleus where they regulate gene expression by activating transcription factors such as c-JUN, c-FOS, ELK, and ETS, and promote the expression of genes involved in cell proliferation and survival, such as Cyclin D1 [64]. Additionally, ERK phosphorylates other targets like BIM, MCL, and RSK, affecting various cellular processes. MEK and ERK can be inhibited by specific inhibitors (MEKi and ERKi, respectively). This pathway is crucial for cell survival and proliferation, and its components represent significant targets for therapeutic intervention in diseases characterized by uncontrolled cell growth, such as cancer (Fig 10) [65].

Future research should focus on optimizing the identified phytochemicals for better bioavailability and potency, as well as exploring their combinatorial effects with existing cancer therapies [66]. The Protein-Protein interaction performed by the String Database showed the MAPK/ERK cascade pathway interaction. The results of this PPI networking indicated that several other proteins are indeed co-expressed and co-activated with ERK, highlighting their significance in maintaining the pathway's functionality. These findings are crucial as they identify potential secondary targets and mechanisms that could be explored for developing more comprehensive anticancer therapies [67]. Most genetic and epigenetic alterations contribute to the dysregulation of various signal transduction pathways in cancer [68]. In conclusion, this study identifies several promising phytochemical inhibitors of the MAPK/ERK pathway, providing a foundation for future research and development of novel anticancer therapies [69, 70]. The insights gained from this study emphasize the potential of integrating natural products into cancer treatment strategies, potentially leading to more effective and less toxic therapeutic options.

## Conclusion

This study highlights the potential of certain phytochemicals, such as luteolin (CID 5280445), hispidulin (CID 5281628), and isorhamnetin (CID 5281654), as potent inhibitors of the MAPK/ERK signaling pathway, which is crucial in RAS-driven cancers. Through molecular docking and simulations, these compounds demonstrated significant binding affinities to ERK proteins, particularly at the allosteric site, indicating their ability to inhibit ERK activity. The stability of these phytochemical-protein complexes was confirmed by their consistent radius of gyration values. The study evaluated the stability of protein-ligand complexes, supporting the role of ERK1/2 in cancer therapy. Complexes 3, 7, and 8 which are luteolin, hispidulin, and isorhamnetin, respectively showed stable binding over a 200-nanosecond simulation, underscoring the potential of targeting ERK1/2. Further studies are needed to confirm the preclinical and clinical potential of these natural compounds targeting the ERK pathway which offers a promising approach for developing effective and less toxic anticancer therapies.

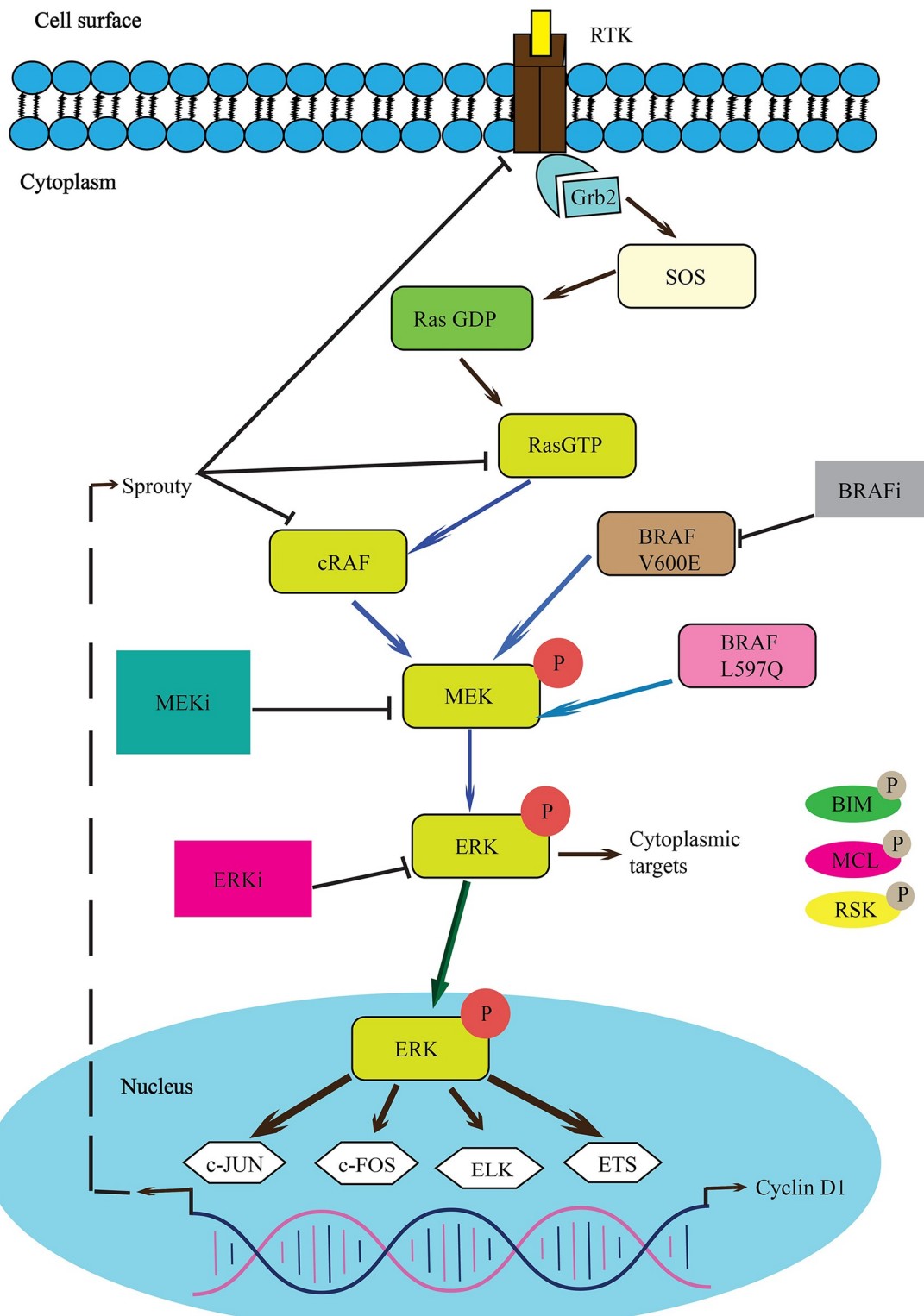

**Fig 10. An illustration of the pathway involved in various types of cancer.** The inhibition of the downstream protein can be a potential for the cancer therapeutic study.

## Supporting information

**S1 Fig. A 200-nanosecond simulation is conducted to measure the root mean square deviation (RMSD).** A) RMSD of Complex 1 B) RMSD of Complex 2, C) RMSD of Complex 4, D) RMSD of Complex 5, E) RMSD of Complex 6, and F) RMSD of Complex 9. The root means square deviation (RMSD) between the ligand and protein exhibits temporal inconstancy, thereby ensuring instability. Complex 1 showed stability for a certain time and then the ligand was out of the protein contact at 75 to 100ns, indicating poor stability.
(TIF)

**S2 Fig. A 200-nanosecond simulation is conducted to measure the root mean square fluctuation (RMSF).** A) RMSF of Complex 1 B) RMSF of Complex 2, C) RMSF of Complex 4, D) RMSF of Complex 5, E) RMSF of Complex 6, and F) RMSF of Complex 9. The interpretation of the results is justified by Several significant fluctuations. The fluctuation primarily arises when the ligand interacts with the protein residues. All the complexes exhibit several significant fluctuations not more than 0.4nm.
(TIF)

**S3 Fig. The results of a 200ns simulation of the SASA computation for the other complexes.** Protein and the control SASA are displayed in main Fig 3. The SASA calculation was between 160 to 195 $nm^2$. The solvent-accessible surface area (SASA) measurements of the amino acid residues at the C-terminus of one protein are found to be lower than those at the N-terminus of a different protein, indicating a higher degree of hydrophobicity and compactness in the free end amino acid residues of the former protein in comparison to the latter.
(TIF)

**S4 Fig. The radius of gyration in picoseconds was utilized to investigate the compactness and stability of the ligands using the radius of gyration (Rg).** It was noted that the protein had the lowest Rg value, ranging from 2.125 to 2.275 nm, as expected due to its higher compactness compared to other substances.
(TIF)

**S1 Table. The Active side residues.**
(PDF)

**S2 Table. ADMET profiling of sorted compounds.**
(PDF)

**S3 Table. Triplet docking scores (autodock vina and pyrx) with an average score in Kcal/ mol unit.**
(PDF)

**S1 File. Name of the identified 351 compounds.**
(XLSX)

**S2 File. The AMDET profile results of 340 retrieved compounds.**
(XLSX)

**S3 File. Decoy screening of 50 decoy molecules for each compound.**
(XLSX)

## Acknowledgments

The authors acknowledge the logistic support and laboratory facilities of the Department of Biochemistry and Molecular Biology, Shahjalal University of Science and Technology, Sylhet, Bangladesh.

## Author Contributions

**Conceptualization:** Ajit Ghosh.

**Data curation:** Mahir Azmal, Jibon Kumar Paul.

**Formal analysis:** Mahir Azmal, Jibon Kumar Paul, Fatema Sultana Prima, Omar Faruk Talukder.

**Funding acquisition:** Ajit Ghosh.

**Investigation:** Mahir Azmal.

**Methodology:** Mahir Azmal, Fatema Sultana Prima, Omar Faruk Talukder.

**Project administration:** Ajit Ghosh.

**Software:** Jibon Kumar Paul.

**Supervision:** Ajit Ghosh.

**Validation:** Mahir Azmal.

**Visualization:** Mahir Azmal.

**Writing – original draft:** Mahir Azmal.

**Writing – review & editing:** Ajit Ghosh.

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
