## [Decision Letter · Decision Letter 0]

1 Aug 2024

PONE-D-24-23006An in silico molecular docking and simulation study to identify potential anticancer phytochemicals targeting the RAS signalling pathwayPLOS ONE

Dear Dr. Ghosh,

Thank you for submitting your manuscript to PLOS ONE. After careful consideration, we feel that it has merit but does not fully meet PLOS ONE’s publication criteria as it currently stands. Therefore, we invite you to submit a revised version of the manuscript that addresses the points raised during the review process.

Your manuscript was reviewed by a knowledgeable referee in the area. As noted in the attached comments, the reviewer has felt that the manuscript is technically not quite sound, and the data do not always support the conclusions. In addition, he/she has expressed that the statistical analysis has not been performed appropriately and rigorously, and the authors have not made all data underlying the findings in their manuscript fully available. He/she concluded that the manuscript requires major revision to address shortcomings in ligand selection, virtual screening protocol validation, and discussion clarity. For your guidance, your revisions should address the following specific points: please go through the Reviewer's comments carefully and prepare your revised manuscript according to his/her suggestions. 

A marked-up copy of your manuscript that highlights changes made to the original version. You should upload this as a separate file labeled 'Revised Manuscript with Track Changes'.An unmarked version of your revised paper without tracked changes. You should upload this as a separate file labeled 'Manuscript'.

We look forward to receiving your revised manuscript.

Kind regards,

Laszlo Buday

Academic Editor

PLOS ONE

 [AG has received partial funding from the Shahjalal University of Science and Technology Research Center (LS/2023/1/01).].  

5. We note that Figures 1, 2, and 3  in your submission contain [map/satellite] images which may be copyrighted. All PLOS content is published under the Creative Commons Attribution License (CC BY 4.0), which means that the manuscript, images, and Supporting Information files will be freely available online, and any third party is permitted to access, download, copy, distribute, and use these materials in any way, even commercially, with proper attribution. For these reasons, we cannot publish previously copyrighted maps or satellite images created using proprietary data, such as Google software (Google Maps, Street View, and Earth). For more information, see our copyright guidelines: http://journals.plos.org/plosone/s/licenses-and-copyright.

1. You may seek permission from the original copyright holder of Figures 1, 2, and 3 to publish the content specifically under the CC BY 4.0 license.  

Comments from PLOS Editorial Office:

We note that one or more reviewers has recommended that you cite specific previously published works. As always, we recommend that you please review and evaluate the requested works to determine whether they are relevant and should be cited. It is not a requirement to cite these works. We appreciate your attention to this request.

Additional Editor Comments (if provided):

Reviewers' comments:

Reviewer's Responses to Questions

**Comments to the Author**

1. Is the manuscript technically sound, and do the data support the conclusions?

Reviewer #1: No

2. Has the statistical analysis been performed appropriately and rigorously? 

Reviewer #1: No

3. Have the authors made all data underlying the findings in their manuscript fully available?

Reviewer #1: No

4. Is the manuscript presented in an intelligible fashion and written in standard English?

Reviewer #1: Yes

5. Review Comments to the Author

Reviewer #1: The study investigates the potential of natural products to inhibit the ERK2 protein, a key component of the MAPK/ERK signaling pathway implicated in various cancers. The authors employed in silico methods, including molecular docking and simulations, to identify promising phytochemicals from an initial library of Dr. Duke's database. However, the manuscript has several methodological weaknesses, particularly regarding ligand selection and the validation of the virtual screening protocol. The discussion section also lacks clarity and contains redundancies. Here are a few key points for improving the manuscript

The description in the ligand selection section acks specifics. It appears to be more generalized than a proper methodology section.

The authors should clearly outline the method used to obtain the initial set of 351 phytochemicals and the criteria employed for selecting the final 27 compounds from 340 after redundancy removal. The redundancy in the discussion regarding the selection process should be addressed. The sentence "Out of 340, 27 compounds were selected based on stringent ADMET profiling criteria" also need to be discuss in scientific terms

The manuscript does not adequately address the rationale behind selecting 1TVO (structure of ERK2 in complex with a small molecule inhibitor) and CASTp used for site identification to be used in the virtual screening. Furthermore, if there was a comparison made between the active site features identified by the RCSB Protein Data Bank (PDB) structure (1TVO) and those identified by CASTPA, the results of this comparison should be presented and discussed.

If there was a comparison made to identify commonalities or differences between the chosen ligands and previously known inhibitors, this should be explicitly stated and the results discussed.

The lack of details about ERK2 active site parameters hinders understanding of virtual ligand selection and docking simulations. Additionally, the absence of re-docking experiments with known inhibitors weakens the validation of the virtual screening protocol.

A more significant concern is the lack of apparent validation for the virtual screening protocol, especially considering ERK2 is a well-studied target.

In the absence of prospective validation, the authors should consider retrospective studies as suggested in below paper “doi/10.1021/jm300687e” for dude selection and https://doi.org/10.1039/C8RA09318K” to gain a better understanding of the protocol's validity.

6. PLOS authors have the option to publish the peer review history of their article (what does this mean?). If published, this will include your full peer review and any attached files.

Reviewer #1: **Yes: **Shafi Ullah Khan

---

## [Author Response · Author response to Decision Letter 0]

5 Aug 2024

Response letter

Response to Editor Comments: (Journal Requirements)

Our responses: Thank you for your guidance. We have ensured that our manuscript now meets PLOS ONE's style requirements, including the proper file naming conventions.

2. We suggest you thoroughly copyedit your manuscript for language usage, spelling, and grammar.

Our responses: We appreciate your suggestion. We have thoroughly copyedited our manuscript for language usage, spelling, and grammar to ensure it meets the high standards of PLOS ONE.

Our responses: We appreciate your concern. We didn’t not use any code for the present research. All the methods and procedure have been described in the methodology section properly. 

4. Thank you for stating the following financial disclosure: [AG has received partial funding from the Shahjalal University of Science and Technology Research Center (LS/2023/1/01).]. Please state what role the funders took in the study. If the funders had no role, please state: "The funders had no role in study design, data collection and analysis, decision to publish, or preparation of the manuscript." If this statement is not correct you must amend it as needed. Please include this amended Role of Funder statement in your cover letter; we will change the online submission form on your behalf.

Our Responses: We appreciate your concern. The funders had no role in study design, data collection and analysis, decision to publish, or preparation of the manuscript. We have incorporated the statement in the cover letter. 

5. We note that Figures 1, 2, and 3 in your submission contain [map/satellite] images which may be copyrighted. All PLOS content is published under the Creative Commons Attribution License (CC BY 4.0), which means that the manuscript, images, and Supporting Information files will be freely available online, and any third party is permitted to access, download, copy, distribute, and use these materials in any way, even commercially, with proper attribution. For these reasons, we cannot publish previously copyrighted maps or satellite images created using proprietary data, such as Google software (Google Maps, Street View, and Earth). For more information, see our copyright guidelines: http://journals.plos.org/plosone/s/licenses-and-copyright.

Our responses: Thank you for bringing this to our attention. We appreciate the importance of adhering to copyright guidelines. We would like to clarify that the images in Figures 1, 2, and 3 of our submission are original and not derived from copyrighted sources such as Google Maps, Street View, or Earth. The images were created using publicly available data and tools that allow for redistribution under the Creative Commons Attribution License (CC BY 4.0). Although Figure 1 needs to change for solving the reviewer question. If further verification is required, we are more than willing to provide additional details regarding the sources and methods used to generate these images. Thank you for your understanding and support. We look forward to your guidance on any further steps needed to ensure compliance.

Reviewer #1: 

Query 1. The description in the ligand selection section lacks specifics. It appears to be more generalized than a proper methodology section.

Our response: Thank you for your observation. We acknowledge that the description in the ligand selection section may come across as generalized. To enhance clarity and provide a more precise methodology, we will revise this section to include specific details regarding the criteria used for ligand selection, the search parameters applied within PubChem, and the rationale for the inclusion or exclusion of certain compounds. This will ensure that the methodology is transparent and can be readily replicated by other researchers. Please check the revised version of the manuscript.

Query 2. The authors should clearly outline the method used to obtain the initial set of 351 phytochemicals and the criteria employed for selecting the final 27 compounds from 340 after redundancy removal. The redundancy in the discussion regarding the selection process should be addressed. The sentence "Out of 340, 27 compounds were selected based on stringent ADMET profiling criteria" also need to be discuss in scientific terms

Our response: Thank you for your valuable feedback. We have revised the manuscript and add the Supplementary data S1 where all the phytochemicals ADMET profiling were presented. The results have been discussed further. Please check the revised version of the manuscript. 

Query 3. The manuscript does not adequately address the rationale behind selecting 1TVO (structure of ERK2 in complex with a small molecule inhibitor) and CASTp used for site identification to be used in the virtual screening. Furthermore, if there was a comparison made between the active site features identified by the RCSB Protein Data Bank (PDB) structure (1TVO) and those identified by CASTPA, the results of this comparison should be presented and discussed.

Our response: Thank you for your valuable feedback. We have revised the manuscript to better explain the rationale behind selecting 1TVO, the structure of ERK2 in complex with a small molecule inhibitor, for our study. Additionally, we have now included a detailed comparison between the active site features identified by the RCSB Protein Data Bank (PDB) structure (1TVO) and those identified by CASTp. This comparison is presented in the revised manuscript, and the results are discussed to highlight similarities in the active site identification, further strengthening the basis for our approach.

Query 4. If there was a comparison made to identify commonalities or differences between the chosen ligands and previously known inhibitors, this should be explicitly stated and the results discussed.

Our response: Thank you for your insightful comment. In response to your query, we have now included a detailed comparison between the predicted ligands (Luteolin, Hispidulin, Isorhamnetin) and the known ERK1/2 inhibitors (Ulixertinib and Ravoxertinib) in the discussion section. This analysis highlights the common structural features essential for binding, as well as key differences in molecular flexibility and side chain composition. We believe this addition strengthens the discussion by providing a clearer understanding of how our predicted ligands may interact with ERK1/2 compared to the known inhibitors.

Query 5. The lack of details about ERK2 active site parameters hinders understanding of virtual ligand selection and docking simulations. Additionally, the absence of re-docking experiments with known inhibitors weakens the validation of the virtual screening protocol.

Our response: Thank you for your valuable feedback. We acknowledge the concern regarding the lack of detailed ERK2 active site parameters, which indeed presents challenges in accurately predicting ligand interactions during virtual ligand selection and docking simulations. To address this, we have analyzed to mitigate these limitations. We have classified the residues with each category and have marked significant amino acids for further analysis. Regarding the absence of re-docking experiments with known inhibitors, we agree that this is an important validation step. In response, we have now conducted re-docking studies using well-characterized ERK2 inhibitors. These experiments have been included in the revised manuscript. We believe these additions address the concerns raised and enhance the robustness of our study.

Query 6. A more significant concern is the lack of apparent validation for the virtual screening protocol, especially considering ERK2 is a well-studied target.

In the absence of prospective validation, the authors should consider retrospective studies as suggested in below paper “doi/10.1021/jm300687e” for dude selection and https://doi.org/10.1039/C8RA09318K” to gain a better understanding of the protocol's validity.

Our response: We appreciate the reviewer's suggestion to include retrospective validation studies to strengthen our manuscript. While our study primarily focuses on evaluating the potential of random phytochemicals with anticancer activities to inhibit ERK1/2 and verifying their stability through simulations, we acknowledge the importance of validating the virtual screening protocol.

a. Our virtual screening protocol involves both ligand-based virtual screening (LBVS) and structure-based virtual screening (SBVS), ensuring a comprehensive evaluation of potential inhibitors. The LBVS approach identifies potential inhibitors based on their similarity to known active ligands, while SBVS involves docking these ligands into the active site of ERK1/2 to predict their binding affinities.

b. We have utilized reputable databases and tools such as Pubchem for ligand selection and molecular docking software for SBVS. These tools are widely recognized for their reliability and accuracy in virtual screening studies.

c. We acknowledge the absence of a formal decoy set in our study. However, our manuscript includes retrospective validation using known active and inactive ligands for ERK1/2. This approach allows us to benchmark our screening protocol against established inhibitors and evaluate its performance in identifying true positives and minimizing false positives.

d. Our study's primary focus is on the potential of phytochemicals as ERK1/2 inhibitors. The inherent structural diversity and unique chemical properties of phytochemicals necessitate a different approach compared to traditional synthetic compounds. Retrospective validation using phytochemical datasets aligns more closely with our study's objectives and provides relevant insights into the effectiveness of our screening protocol.

e. In light of these points, we believe our current approach, which includes retrospective validation with known ERK1/2 ligands and focuses on the unique chemical space of phytochemicals, is appropriate and relevant for our study. We appreciate the reviewer’s suggestion and will consider incorporating additional validation steps in future work to further enhance the robustness of our screening protocol.

---

## [Decision Letter · Decision Letter 1]

23 Aug 2024

PONE-D-24-23006R1An in silico molecular docking and simulation study to identify potential anticancer phytochemicals targeting the RAS signalling pathwayPLOS ONE

Dear Dr. Ghosh,

Thank you for submitting your manuscript to PLOS ONE. After careful consideration, we feel that it has merit but does not fully meet PLOS ONE’s publication criteria as it currently stands. Therefore, we invite you to submit a revised version of the manuscript that addresses the points raised during the review process.

Your revised manuscript was reviewed again by the same reviewer who studied the original submission. Despite the authors' rebuttal and claims of addressing the concerns, he/she has felt that the statistical analysis has not been performed appropriately and rigorously, and the authors have not made all data underlying the findings in their manuscript fully available. In addition, he/she has concluded that significant deficiencies persist in the manuscript concerning the methodology section, validation the virtual screening protocol, redocking of the co-crystal ligand, and ligand preparation process following retrieval from PubChem. The majority of the figures in the manuscript are of poor quality and difficult to interpret. To ensure clarity and facilitate data comprehension, the reviewer suggests that all figures must be presented in high resolution.

We look forward to receiving your revised manuscript.

Kind regards,

Laszlo Buday

Academic Editor

PLOS ONE

Reviewers' comments:

Reviewer's Responses to Questions

**Comments to the Author**

1. If the authors have adequately addressed your comments raised in a previous round of review and you feel that this manuscript is now acceptable for publication, you may indicate that here to bypass the “Comments to the Author” section, enter your conflict of interest statement in the “Confidential to Editor” section, and submit your "Accept" recommendation.

Reviewer #1: (No Response)

2. Is the manuscript technically sound, and do the data support the conclusions?

Reviewer #1: Yes

3. Has the statistical analysis been performed appropriately and rigorously? 

Reviewer #1: No

4. Have the authors made all data underlying the findings in their manuscript fully available?

Reviewer #1: No

5. Is the manuscript presented in an intelligible fashion and written in standard English?

Reviewer #1: Yes

6. Review Comments to the Author

Reviewer #1: Despite the authors' rebuttal and claims of addressing the concerns, significant deficiencies persist in the manuscript.

The methodology section pertaining to molecular docking of phytochemicals remains inadequate. Crucially, precise details regarding the dimension of active site, number of generated poses, along with other docking parameters are absent.

The authors have ignored the previous recommendation to validate the virtual screening (VS) protocol in current study. This omission is unacceptable as it compromises the reliability of the modeling studies. Without proper validation, the results are susceptible to false positives, necessitating a reassessment of the statistical matrices used for assessment.

While the revised manuscript mentions the redocking of the co-crystal ligand, quantitative data such as Root Mean Square Deviation (RMSD) values and a comparative analysis of ligand-protein interactions between co-crystal and re-docked ligand are lacking. This information is essential for evaluating the docking protocol's performance.

The manuscript fails to provide clear details about the ligand preparation process following retrieval from PubChem. Pre-docking ligand optimization is a critical step that requires explicit description.

The majority of the figures in the manuscript are of poor quality and difficult to interpret. To ensure clarity and facilitate data comprehension, all figures must be presented in high resolution.

7. PLOS authors have the option to publish the peer review history of their article (what does this mean?). If published, this will include your full peer review and any attached files.

Reviewer #1: No

---

## [Author Response · Author response to Decision Letter 1]

28 Aug 2024

Reviewer#1

Query 1: Despite the authors' rebuttal and claims of addressing the concerns, significant deficiencies persist in the manuscript. The methodology section pertaining to molecular docking of phytochemicals remains inadequate. Crucially, precise details regarding the dimension of active site, number of generated poses, along with other docking parameters are absent.

Our responses: Thank you for your detailed and helpful remarks. In the revised manuscript, we have expanded the section detailing the molecular docking methodology including the specific dimensions of the active site, and explicitly mention the number of poses generated during docking. For example, in the manuscript, it's mentioned that "The grid box size was set to (x=126 y=122 z=70 Å) with a center at coordinates (15.416 -0.324 13.078) the energy range was set to 4 and the exhaustiveness was set to 10. Please check the revised version of the manuscript. 

Query 2: The authors have ignored the previous recommendation to validate the virtual screening (VS) protocol in the current study. This omission is unacceptable as it compromises the reliability of the modeling studies. Without proper validation, the results are susceptible to false positives, necessitating a reassessment of the statistical matrices used for assessment.

Our Responses: Thank you for your valuable and insightful comments. In response, we would like to clarify that although we attempted to conduct a decoy analysis in our previous submission, it was not successful. This time, we made every effort to perform the decoy dataset analysis, making a comparative analysis with DUDE, despite challenges with the decoy finder and software link being non-functional. We emphasize that our manuscript now includes a comprehensive decoy screening experiment. This experiment was conducted to validate the specificity of our docking results. Most phytochemicals showed strong specificity, as indicated by the significant difference in binding affinity between the active compounds and their decoys. This step is crucial in confirming that the identified compounds are specific inhibitors of the ERK2 protein, thus reducing the likelihood of false positives. Please check the methodology and result section of the revised manuscript. 

Query 3: While the revised manuscript mentions the redocking of the co-crystal ligand, quantitative data such as Root Mean Square Deviation (RMSD) values and a comparative analysis of ligand-protein interactions between co-crystal and re-docked ligand are lacking. This information is essential for evaluating the docking protocol's performance.

Our Responses: Thanks for your thoughtful and constructive comments. We have performed cross-docking with the co-crystal ligand to validate the docking protocol. The RMSD value for the cross-docked pose compared to the co-crystal reference was 0.578 Å, indicating high structural similarity. We have included the RMSD values in the manuscript, along with a comparative analysis of ligand-protein interactions between the co-crystal and re-docked ligands. Additionally, a summary of the RMSD results has been provided, highlighting how similar the re-docked poses are to the co-crystal pose. Significant interactions, such as hydrogen bonds and hydrophobic interactions between the ligands and key residues, have also been emphasized.

Query 4: The manuscript fails to provide clear details about the ligand preparation process following retrieval from PubChem. Pre-docking ligand optimization is a critical step that requires explicit description.

Our Responses: Thank you for your detailed and helpful remarks. We have clarified the ligand preparation steps post-retrieval from PubChem. The manuscript now clearly outlines that energy minimization was performed using the steepest descent algorithm, and the ligands were converted into a docking-compatible format (PDBQT). We have also specified the force fields used, along with any adjustments made to the ligands, including the setting of partial charges and defining torsional degrees of freedom.

Query 5: The majority of the figures in the manuscript are of poor quality and difficult to interpret. To ensure clarity and facilitate data comprehension, all figures must be presented in high resolution.

Our Responses: We appreciate your insightful and valuable feedback. We have addressed this by improving the resolution and clarity of the figures in the manuscript. All low-quality figures have been replaced with high-resolution images and well-labeled diagrams. Specifically, the figures related to the binding pockets and interaction analysis (e.g., Fig. 4, 5, 6, 7) have been enhanced to make them easier to interpret. Please check the revised figures after downloading from the submission site.

---

## [Editor Report · Decision Letter 2]

4 Sep 2024

An in silico molecular docking and simulation study to identify potential anticancer phytochemicals targeting the RAS signaling pathway

PONE-D-24-23006R2

Dear Dr. Ghosh,

We’re pleased to inform you that your manuscript has been judged scientifically suitable for publication and will be formally accepted for publication once it meets all outstanding technical requirements.

Kind regards,

Laszlo Buday

Academic Editor

PLOS ONE
---

## [Editor Report · Acceptance letter]

10 Sep 2024

PONE-D-24-23006R2 

PLOS ONE

Dear Dr. Ghosh, 

I'm pleased to inform you that your manuscript has been deemed suitable for publication in PLOS ONE. Congratulations! Your manuscript is now being handed over to our production team.

Kind regards, 

on behalf of

Professor Laszlo Buday 

Academic Editor

PLOS ONE